# Contextual and individual factors associated with self-reported tooth loss among adults and elderly residents in rural riverside areas: A cross-sectional household-based survey

Vitor Guilherme Lima de Souza[1‡], Fernando José Herkrath[2,3☯], Luiza Garnelo[2☯], Andréia Coelho Gomes[2☯], Uriel Madureira Lemos[2‡], Rosana Cristina Pereira Parente[2☯], Ana Paula Corrêa de Queiroz Herkrath[1☯]*

1 School of Dentistry, Federal University of Amazonas, Manaus, Amazonas, Brazil, 2 Instituto Leônidas e Maria Deane, Fundação Oswaldo Cruz, Manaus, Amazonas, Brazil, 3 Superior School of Health Siences, State University of Amazonas, Manaus, Amazonas, Brazil

☯ These authors contributed equally to this work.
‡ VGLS and UML also contributed equally to this work.
* anapaulaqueiroz@gmail.com

**Data Availability Statement:** The data that support the findings of this study are openly available in the

## Abstract

### Background

Tooth loss is an oral health condition with high prevalence and negative impact on quality of life. It is the result of the history of oral diseases and their treatment as well as provision of dental care and access to dental services. Socioeconomic characteristics are determinants of tooth loss and living in rural areas is also a risk factor for its incidence.

### Objective

To identify contextual and individual factors associated with self-reported tooth loss among adults and elderly people living in rural riverside areas.

### Methods

A cross-sectional household-based survey was conducted in 2019 with rural riverside communities on the left bank of the Rio Negro River, Manaus, Amazonas. These communities are covered by a fluvial health team and two riverside health teams. Interviews were conducted in a representative random sample of dwellers aged ≥ 18 years, using electronic forms to obtain information on oral health conditions, demographic and socioeconomic characteristics, and use of and access to health services. The outcome was self-reported tooth loss. After the descriptive analysis of the data, a multilevel Poisson regression analysis was performed to estimate the prevalence ratio for the outcome. Variables with p-value ≤0.20 in the bivariate analyses were included in the multiple analysis considering the hierarchy between individual and contextual variables in the multilevel model. Variables with p-value ≤0.10 were kept in the final model and the significance level adopted was 0.05.

Open Science Framework: Herkrath, F. J. (2022, October 11). "Contextual and individual factors associated with self-reported tooth loss among adults and elderly residents in rural riverside areas: a cross-sectional household-based survey". Retrieved from www.osf.io/6sjf9.

**Funding:** This study was funded by Fundação Oswaldo Cruz - PROEP-Labs/ILMD Fiocruz Amazônia, call for proposals 001/2020 (https://amazonia.fiocruz.br), Coordenação de Aperfeiçoamento de Pessoal de Nível Superior - CAPES PDPG Amazônia Legal, call for proposals 013/2020 (https://www.gov.br/capes), and Fundação de Amparo à Pesquisa do Estado do Amazonas -FAPEAM Posgrad (http://www.fapeam.am.gov.br). The funders had no role in study design, data collection and analysis, decision to publish, or preparation of the manuscript.

**Competing interests:** The authors have declared that no competing interests exist.

## Results

603 individuals from 357 households were assessed (mean age 44.1 years). The average number of missing teeth was 11.2 (±11.6); 27.4% of individuals had lost more than 20 teeth (non-functional dentition) and 12.1% were completely edentulous. Contextual characteristic of primary healthcare offered was associated with the outcome. The tooth loss was lower in territories covered by riverside health teams. At individual level, tooth loss was greater in older individuals who had experienced dental pain over the past six months and whose sugar consumption was high. Black or brown individuals, individuals whose household income was higher, those who were on the Bolsa Família cash transfer program, those who consulted a dentist over the past year, and those who reported satisfaction with their teeth/oral health reported less tooth loss.

## Conclusion

Tooth loss was associated with contextual territorial factors related to the healthcare service and individual demographic, behavioral, socioeconomic, and service-related characteristics as well as self-perceived oral health conditions. The findings suggest that actions focused on the oral health of these populations involve not only changes in the healthcare service organization, but also intersectoral policies that contribute to reducing social inequalities.

## Introduction

Tooth loss is one of the main oral health problems. It is a potentially avoidable and complex oral health outcome that reflects the history of dental disease during individual's life course and its treatment. The esthetic and functional impact [1–3], as well as the psychological [4] and social impairments [5] negatively affect the quality of life of individuals [6]. Total tooth loss is the leading cause of disability-adjusted life years due to oral conditions [7].

Although tooth loss has decreased in all age groups in many developed countries, despite significant geographic differences, its prevalence is still high, especially in developing countries [7, 8]. In Brazil, the last oral health survey (SBBrasil 2010) revealed that tooth loss had decreased in adolescents and adults compared to the previous survey, but not among the elderly, whose prevalence of edentulism is just over 50% with an average of 25.4 missing teeth. In adults, the absence of functional dentition (20 natural teeth or more) was observed in approximately 25% of adults and the average of missing teeth declined from 13.5 in 2003 to 7.4 in 2010 [9, 10]. The National Health Survey (NHS), conducted in 2013, showed that total tooth loss affects around 16 million Brazilians [11]. Data from the most recent national survey estimated an 83.3% prevalence of loss of at least one tooth among those aged 30 years or older and 10% of edentulous individuals [12].

The complexity of the tooth loss depicts the endpoint of the most frequent oral diseases, dental caries and periodontal diseases [13], combined with the access and utilization of dental services, the hegemonic curative and mutilating dental care model and the individual health-related behaviors [14–20]. The socioeconomic conditions are the underlying factors that play a significant role in tooth loss [15, 19–22]. These factors involved in the causal pathway of tooth loss are expressed either contextually or individually.

Tooth loss is an expression of social inequalities. It is higher in population groups at the bottom of the socioeconomic hierarchy than those at the top. It is observed both when

individual socioeconomic characteristics (such as family income and education level) or contextual ones (such as the Human Development Index) are considered [20, 22–26]. In Brazil, tooth loss is higher among adult and elderly population whose income and education are lower and who declare themselves black and brown [9–11, 20, 22].

Additionally, living in rural areas can be considered a risk factor for tooth loss [27, 28], as it is to oral diseases as a whole [29, 30]. Data from the Brazilian NHS reveal that tooth loss is more prevalent in individuals who live in rural areas [11]. Contextual characteristics of geographic location contributes to inequalities between rural and urban areas, both in general and oral health. The urban environment can mitigate the negative aspects present in the rural environment, such as geographic barriers, socioeconomic deprivation, and limited access to health services [31–35]. Despite having a strong dependence on public healthcare services [36], the physical network of public and private services is inadequate and health professionals are lacking in rural areas in Brazil [37]. Brazil is a large and unequal country, and the northern region has the worst rates of use of health services in the country [38]. Health policies in the region lack institutional structure, continuity, and sensitivity to regional specificities. In addition to the limited provision and organization of healthcare of services clinics, there are low income, population dispersion, the large geographic distances and other barriers to access typical of the Amazon region [39–41].

Although there is evidence that the rural Brazilian populations have a high rate of dental caries, periodontal diseases [42, 43] and tooth loss [11, 44, 45], data on the oral health of populations in rural areas of the Amazon region are scarce. Two studies have assessed self-reported outcomes: one study compared urban and rural communities in Pará and found that dental caries and periodontal diseases significantly affected the quality of life of individuals from remote communities [46]; another study revealed that the prevalence of oral health impacts on the quality of life was high in rural communities in Amazonas [47].

While the scientific basis for oral health actions apply to all populations, social and geographic factors might make them peculiar in rural areas. As the place contextualizes health, a deep understanding of the epidemiological profile of tooth loss together with the factors associated with it in rural areas can subsidize effective planning of oral health actions and services taking into account the geographic and social specificities that remoteness brings with it, particularly in the Amazon region. Thus, the aim of the study was to evaluate the association of contextual and individual factors with self-reported tooth loss among adults and elderly people living in rural riverside areas.

## Methods

A household-based cross-sectional survey was conducted from March to July 2019 in rural riverside localities along the left bank of Negro River, Manaus, Amazonas, Brazil. The study population included five areas covered by a fluvial family health team (Apuaú, Mipindiaú, Cuieras, Santa Maria and Costa do Arara) and two areas covered by a riverside family health team each (Nossa Senhora do Livramento and Nossa Senhora de Fátima), encompassing seven territories (Fig 1). There were no health professionals based in the territory, except for community health workers (CHW) who are part of the health teams. In the fluvial teams, the other professionals perform most of their functions in a mobile fluvial health unit (vessel), visiting the localities intermittently on monthly trips. In riverside teams, most of the functions are carried out in health units built/located in the communities, which are closer to the urban area, with daily displacement of professionals to the health establishment by river.

Stratified random sampling was performed based on the number of individuals and households in each community, totaling 3811 people residing in 1350 households as reported by the

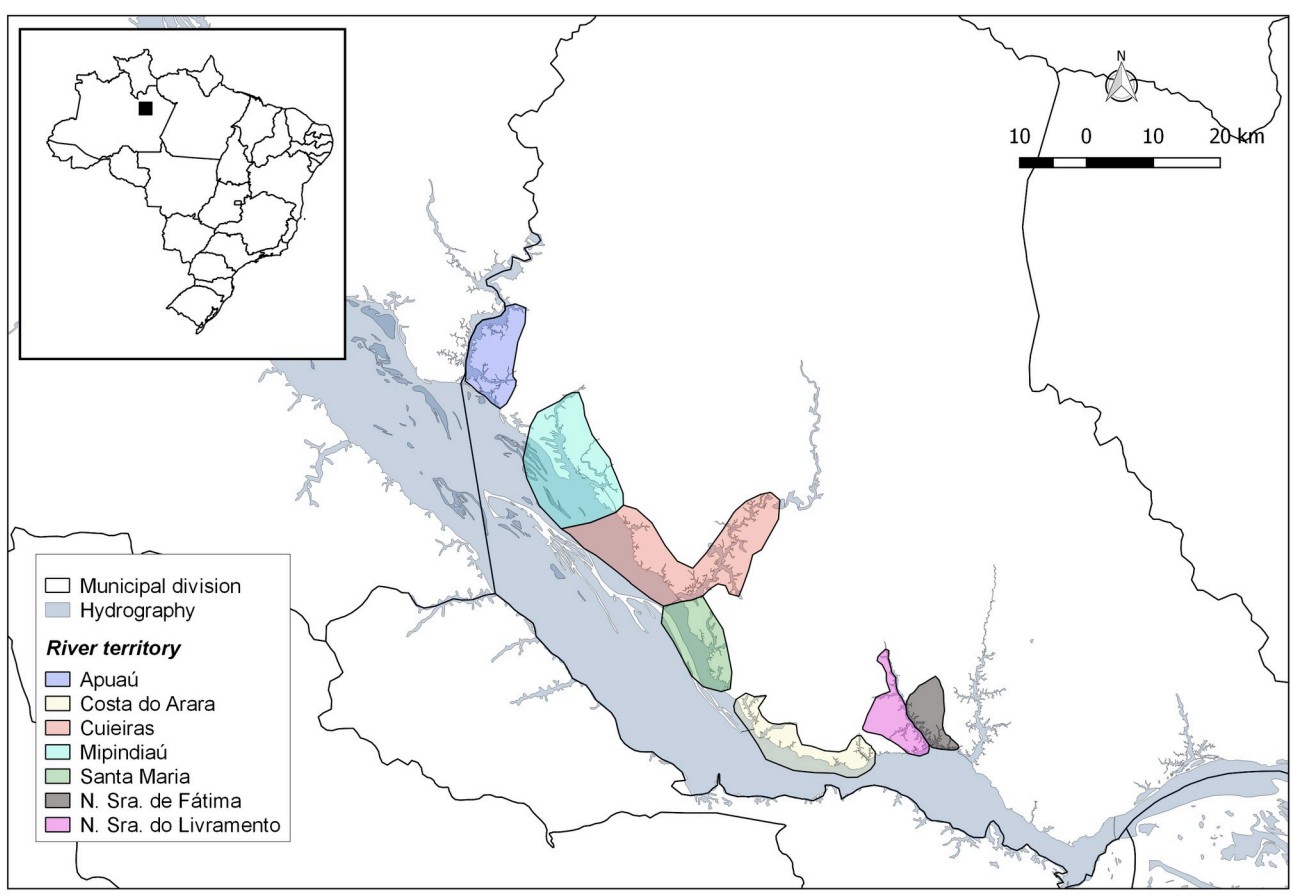

DATUM: SIRGAS 2000 (EPSG: 4674)      Update date: 04/11/2021

**Fig 1. Territories covered by the fluvial and riverside health teams.** Municipal division data were obtained from Instituto Brasileiro de Geografia e Estatística, IBGE (public domain): https://www.ibge.gov.br/en/geosciences/territorial-organization/territorial-meshes/ and hydrography data were obtained from Instituto Nacional de Pesquisas Espaciais, INPE (public domain): http://www.dpi.inpe.br/Ambdata/English/.

CHW. The sample size calculation considered a prevalence of 50% of the health outcomes of interest and precision of 0.05, in addition to 10% of possible losses or refusals, adjusted for the finite population, and was carried out in a representative way for adults of both sexes and the elderly, resulting in 753 individuals aged $\geq$ 18 years. Considering the probability of finding individuals from each group within households (average of 2.215 adults per household), a number of 340 households were estimated to be visited in the study, distributed throughout the territory according to the stratification of the sample by the population size of the forty rural localities comprised in the study.

The households in each community were included by systematic random selection. The sampling interval was determined by the total number of households in each locality divided by the number of households sampled. To select the first household within the first ones located in the initial interval (or the only one selected in the set of households) a random number generator program or table was used, and the ordering of households followed the natural order along the rivers, or the organization of houses and circulation routes in larger rural communities. The random generator program was also used to choose the resident selected in the household when there was more than one individual in each group of interest. Thus, randomness was ensured in all stages of selection.

The interview was conducted using questionnaires developed in the Research Electronic Data Capture (REDCap), an open-source application to create and manage surveys and databases. To make it feasible to cover the entire population dispersed in the study territory, twenty examiners were responsible for data collection. Examiners stayed in most locations for an extended period of time throughout the fieldwork days to minimize possible losses. After 40 hours of theoretical and practical training, a pilot study was conducted in two rural communities, from territories other than those included in the main study. A questionnaire organized in seven thematic sections was used to collect information directly from the residents by trained interviewers, including the characterization of socioeconomic status, health conditions, and use of and access to health services, with an average application time of 60 minutes per household. The outcome of interest for this study was self-reported tooth loss that was obtained through two questions reproduced from the NHS: 1) As for your upper teeth, how many teeth have you lost? and 2) As for your bottom teeth, how many teeth have you lost? The number of missing teeth was the sum of the numerical answers to the two questions, ranging from 0 to 32.

Independent variables were selected according to the theoretical background presented on the determinants of tooth loss. Contextual variables included for each territory were per capita income, average household income, poverty (family income less than US$ 5.50 per day) and extreme poverty (family income less than US$ 1.90 per day) rates, Gini index of per capita income and household income. These continuous numeric variables were calculated based on information from the households evaluated in the territories. Characteristic of primary healthcare was also assessed at a contextual level using a nominal categorical variable, including: 1) locations covered by fluvial health teams (FHT) and CHW, with health unit in the community, 2) locations covered by FHT and CHW, without health unit in the community, 3) locations only covered by CHW, and 4) locations covered by riverside health teams (RHT).

Individual characteristics were sex (male/female), age (discrete numeric variable), race/skin color (white/black/brown/Asiatic/indigenous), household income (continuous numeric variable), registration (yes/no) in the Bolsa Família program (Brazilian conditional cash transfer program for low-income families), occupation (yes/no), monthly sugar consumption of family, time since last dental appointment, dental pain over the past six months (yes/no) and satisfaction with teeth/oral health. Time since last dental appointment was evaluated according to the categories: over the past 12 months, more than 1 year up to 2 years ago, more than 2 years up to 3 years ago, more than 3 years ago, and never have been to the dentist. Satisfaction was assessed using a 5-point Likert scale, ranging from very satisfied to very dissatisfied. Sugar consumption was evaluated in kilograms (continuous numeric variable) through the availability inferred by the frequency of monthly purchases of sugary foods at home.

The data collected in the study were directly exported from REDCap to the database files of the Stata program. Initially, a descriptive analysis of the data was performed. In addition to the main outcome of interest, the number of missing teeth, dichotomous outcomes of total tooth loss (complete edentulism), severe tooth loss (up to 8 natural teeth) and non-functional dentition (less than 20 natural teeth) were also described. These conditions are not mutually exclusive. Then, Poisson regression analysis was performed to evaluate the variables associated with the number of missing teeth (count outcome), estimating the rate ratios and respective 95% confidence intervals. Poisson regression coefficients (β) can be interpreted as the log of the rate ratio. Thus, the rate ratio was obtained by exponentiating the Poisson regression coefficient. In the analyses, rate ratios represent the expected count outcome for *X+1* divided by the expected count outcome for *X*. A multilevel modeling analysis was carried out to include the hierarchical structure or grouping of the study population in the respective territories (Fig 2). Variables with p-values lower than or equal to 0.20 in the bivariate analyses were included in

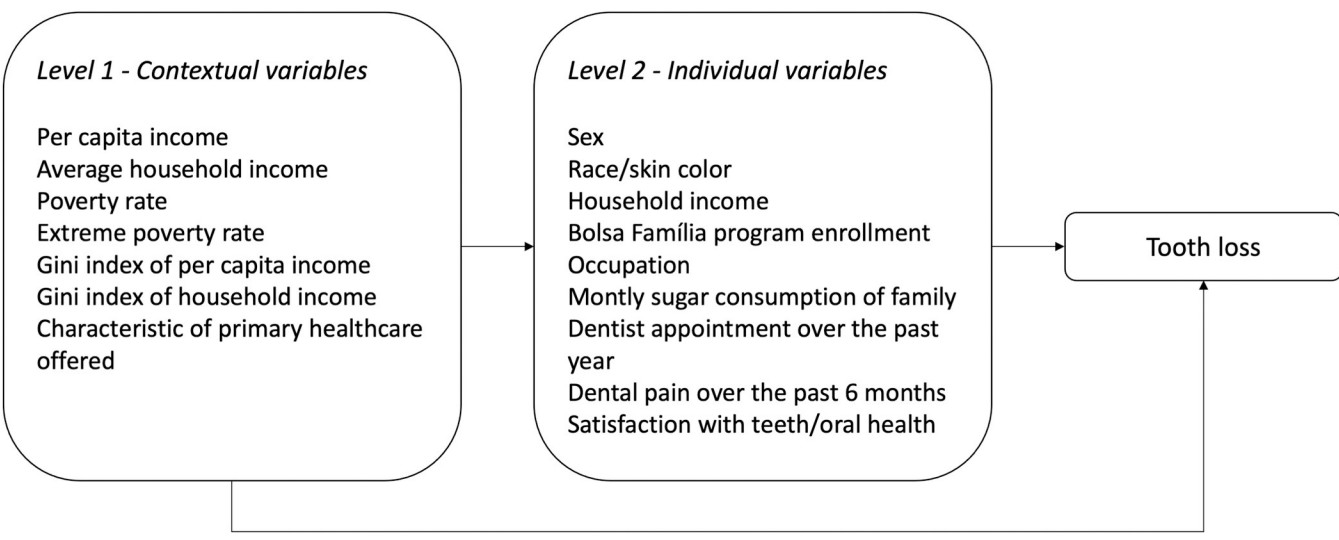

**Fig 2. Hypothesized hierarchical analytical model for tooth loss in the rural riverside population.**

the multiple analysis considering the hierarchy between individual and contextual variables in the model. The first model included only the contextual variables. The individual variables were included in the second model. Variables with p-values lower than or equal to 0.10 when included into the hierarchical model were maintained in the final model. The significance level adopted was 0.05.

This study is part of a broader project that aimed to evaluate the living and health conditions and access to health services of rural riverside populations. The study was approved by the Research Ethics Committee (CAAE No. 57706316.9.0000.0005) and written informed consent was obtained from individuals to participate in the study.

## Results

A total of 603 individuals residing in 357 households distributed throughout the seven territories along the Negro River were assessed. The mean age of the individuals was 44.1, ranging from 18 to 90.7 years, and 50.9% were female. The mean number of self-reported missing teeth was 11.2 (±11.6), ranging from no missing teeth (9.9%, n = 60) to total tooth loss (12.1%, n = 73). A total of 27.4% of individuals had less than 20 teeth (non-functional dentition) and 19.6% of individuals presented severe tooth loss (less than 8 natural teeth). Table 1 shows the contextual and individual characteristics of the study participants.

Prior to the multilevel analyses, the null model was adjusted to verify contextual effects. The null model showed a statistically significant random effect [Var = 0.017, 95% confidence interval (CI) = 0.006–0.054], revealing significant differences between the territories. The likelihood-ratio test of the null model versus the OLS baseline model was also significant (p<0.001), indicating the need for multilevel modeling since the estimates of the model differed significantly from the estimates obtained from the model with no level structures. Table 2 shows the results of the bivariate analyses.

Table 3 shows the adjusted rate ratios, indicating the association of contextual characteristic of primary healthcare offered with self-reported number of missing teeth. Locations covered by riverine teams presented lower tooth loss than those covered by fluvial teams with CHW in a health facility in the territory. At individual level, a higher number of missing teeth was found in older individuals, those who consumed more sugar at home, who had experienced

**Table 1. Contextual and individual characteristics of adults and elderly residents in rural riverside localities included in the study, Negro River, Manaus, Amazonas, 2019.**

| Variable | n (%) / mean (±SD) |
| --- | --- |
| *Contextual characteristics* | |
| Per capita income (BRL) | 335.34 (±370.77) |
| Average household income (BRL) | 1073.28 (±121.58) |
| Poverty rate | 50.3 (±15.6) |
| Extreme poverty rate | 29.5 (±10.6) |
| Gini index of per capita income | 0.495 (±0.064) |
| Gini index of household income | 0.393 (±0.043) |
| *Individual characteristics* | |
| Sex | |
| Male | 296 (49.1%) |
| Female | 307 (50.9%) |
| Age | 44.1 (±17.9) |
| Race/skin color | |
| White | 62 (10.3%) |
| Black | 42 (7.0%) |
| Brown | 441 (73.1%) |
| Indigenous | 52 (8.6%) |
| Ignored | 6 (1.0%) |
| Household income (BRL) | 1107.62 (±872.03) |
| Bolsa Família program | |
| No | 334 (55.4%) |
| Yes | 269 (44.6%) |
| Working residents | |
| No | 101 (16.8%) |
| Yes | 502 (83.3%) |
| Monthly sugar consumption of family (kg) | 6.8 (±5.0) |
| Last dentist appointment | |
| Never | 18 (3.0%) |
| Over the past 12 months | 296 (49.1%) |
| More than 1 year up to 2 years ago | 98 (16.2%) |
| More than 2 years up to 3 years ago | 45 (7.5%) |
| More than 3 years ago | 146 (24.2%) |
| Dental pain over the past 6 months | |
| No | 478 (79.3%) |
| Yes | 125 (20.7%) |
| Satisfaction with teeth/oral health | |
| Does not know/did not answer | 3 (0.5%) |
| Very satisfied | 45 (7.4%) |
| Satisfied | 326 (54.1%) |
| Neither satisfied nor dissatisfied | 76 (12.6%) |
| Dissatisfied | 136 (22.5%) |
| Very dissatisfied | 17 (2.9%) |

BRL, Brazilian Real

**Table 2. Unadjusted rate ratios for the contextual and individual variables and the self-reported number of missing teeth, Negro River, Manaus, Amazonas, 2019.**

| Variable | exp (β) | 95%CI | p-value |
|---|---|---|---|
| *Contextual characteristics* | | | |
| Per capita income | 1.00 | 1.00–1.00 | <0.001 |
| Average household income | 1.00 | 1.00–1.00 | 0.142 |
| Poverty rate | 0.71 | 0.40–1.26 | 0.243 |
| Extreme poverty rate | 0.94 | 0.34–2.62 | 0.908 |
| Gini index of per capita income | 1.36 | 0.28–6.72 | 0.706 |
| Gini index of household income | 2.07 | 0.33–13.00 | 0.437 |
| Characteristic of primary healthcare offered (ref.: locations covered by FHT and CHW, with health unit in the community) | | | |
| Locations covered by FHT and CHW, without health unit in the community | 0.97 | 0.89–1.05 | 0.398 |
| Locations only covered by CHW | 1.17 | 1.09–1.25 | <0.001 |
| Locations covered by RHT | 1.16 | 0.95–1.42 | 0.145 |
| *Individual characteristics* | | | |
| Sex (ref.: male) | | | |
| Female | 0.98 | 0.94–1.03 | 0.460 |
| Age | 1.04 | 1.04–1.04 | <0.001 |
| Race/skin color (ref.: white) | | | |
| Black | 0.86 | 0.77–0.95 | 0.005 |
| Brown | 0.67 | 0.63–0.72 | <0.001 |
| Indigenous | 0.95 | 0.86–1.05 | 0.296 |
| Household income (log[BRL]) | 1.29 | 1.25–1.33 | <0.001 |
| Bolsa Família program (ref.: no) | | | |
| Yes | 0.43 | 0.40–0.45 | <0.001 |
| Working residents (ref.: no) | | | |
| Yes | 0.47 | 0.45–0.50 | <0.001 |
| Montly sugar consumption of family (10kg) | 1.20 | 1.15–1.26 | <0.001 |
| Dentist appointment over the past year (ref.: no) | | | |
| Yes | 0.67 | 0.64–0.71 | <0.001 |
| Dental pain over the past 6 months (ref.: no) | | | |
| Yes | 0.86 | 0.81–0.92 | <0.001 |
| Satisfaction with teeth/oral health (ref.: very satisfied / satisfied) | | | |

*(Continued)*

**Table 2.** (Continued)

| Variable | exp (β) | 95%CI | p-value |
|---|---|---|---|
| Very dissatisfied/ dissatisfied/ neither satisfied nor dissatisfied | 1.17 | 1.11–1.23 | <0.001 |

FHT, fluvial health team

RHT, riverside health team

CHW, community health worker

BRL, Brazilian Real

dental pain over the past six months, and those who reported worse satisfaction with their teeth/oral health. Individuals who reported less tooth loss were those who declared their skin color as black or brown, those who benefit from the Bolsa Família program, those with a higher household income, and those who attended dental appointments over the past year.

## Discussion

Tooth loss has been gaining attention as an indicator for monitoring oral health. Socioeconomic and demographic factors and characteristics related to health service can play a role in this complex outcome. In the present study, which covered rural areas along the Negro River, in Manaus, the RHT were a protective contextual factor against tooth loss. As for individual characteristics, older individuals, who consumed more sugar at home, had experienced dental pain in the past six months, and who reported worse satisfaction with their teeth/oral health, presented a high number of self-reported missing teeth. Black or brown individuals reported less tooth loss as well as those whose benefit from the income transfer program, those with a higher household income, and those who attended dental appointments over the past year.

The mean number of missing teeth reported was high. Few individuals had not lost any teeth and approximately 27% presented non-functional dentition. Rural populations of elderly living the southeast and south of Brazil presented around 50% of edentulism [2, 48]. The last representative epidemiological survey in Brazil was carried out in 2010 in urban areas in the northern inland region and in state capitals. The survey reported that the average of missing teeth in the northern inland region for the age groups 15–19 years, 35–44 years, 65–74 years was 1.2, 11.3 and 27.4, respectively. In the capital of the state of Amazonas, the average of missing teeth was 0.6, 11.0 and 26.2, respectively. This study also showed that the prevalence of tooth loss (ages 15–19), presence of less than 21 natural teeth (ages 35–44) and edentulism (ages 65–74) was 44%, 40% and 56%, respectively [10].

The RHT figured as a contextual protective factor against tooth loss in rural areas along the Negro River. Although the literature suggests that locations with greater coverage of the service tend to present greater tooth loss, especially due to the characteristics of the healthcare model [49], the intermittent presence of fluvial health teams in the territories under their responsibility might mean greater difficulty carrying out health promotion actions when compared to the areas covered by riverside health teams. Garnelo et al. [50] suggest that an expansion of the role of community health agents in these territories could contribute to overcoming the limitations of the current healthcare model due to the short stay of fluvial health teams in each community as well as recognizing the specificities of rural riverside populations when organizing health services to overcome their social exclusion and invisibility. This would contribute to a more rational use of the limited human, financial and structural resources characteristic of the northern region of Brazil and also to a greater resoluteness of health care [39, 40].

**Table 3. Final hierarchical model with adjusted rate ratios for the self-reported number of missing teeth, Negro River, Manaus, Amazonas, 2019.**

| Variable | exp(β) | 95% CI | p-value |
|---|---|---|---|
| *Contextual characteristics* | | | |
| Per capita income | 1.00 | 1.00–1.00 | 0.188 |
| Characteristic of primary healthcare offered (ref.: locations covered by FHT and CHW, with health unit in the community) | | | |
| Locations covered by FHT and CHW, without health unit in the community | 0.96 | 0.88–1.04 | 0.274 |
| Locations only covered by CHW | 1.03 | 0.95–1.11 | 0.472 |
| Locations covered by RHT | 0.88 | 0.79–0.99 | 0.038 |
| *Individual characteristics* | | | |
| Age | 1.04 | 1.04–1.04 | <0.001 |
| Race/skin color (ref.: white) | | | |
| Black | 0.82 | 0.74–0.91 | <0.001 |
| Brown | 0.85 | 0.79–0.91 | <0.001 |
| Indigenous | 1.00 | 0.91–1.10 | 0.989 |
| Total household income (log[BRL]) | 0.95 | 0.91–0.98 | 0.010 |
| Bolsa Família program (ref.: no) | | | |
| Yes | 0.86 | 0.80–0.92 | <0.001 |
| Mantly sugar consumption of Family (10kg) | 1.08 | 1.02–1.14 | 0.004 |
| Dentist appointment over the past year (ref.: no) | | | |
| Yes | 0.90 | 0.85–0.95 | <0.001 |
| Dental pain over the past 6 months (ref.: no) | | | |
| Yes | 1.09 | 1.02–1.16 | 0.012 |
| Satisfaction with teeth/oral health (ref.: very satisfied / satisfied) | | | |
| Very dissatisfied/ dissatisfied/ neither satisfied nor dissatisfied | 1.10 | 1.05–1.16 | <0.001 |

FHT, fluvial health team

RHT, riverside health team

CHW, community health worker

BLR, Brazilian Real

At individual level, age was associated with self-reported tooth loss, as expected. Considering the cumulative effects of oral diseases and, consequently, of tooth loss, the prevalence of tooth loss depends on age, as other studies with national and international data have shown. The main causes of tooth loss are related to chronic and cumulative effects of dental caries and periodontal diseases [8, 10, 13, 16, 51–53]. Other hypotheses that could explain the increase in tooth loss with aging are related to dental service and not always convergent. One of these hypotheses is related to the limited use of dental services [54, 55]. It is also possible that dental

treatment preferences and expectations of older patients and dentists change with aging and the lack of social policies to protect the elderly people can affect their health [54, 56, 57]. On the other hand, the increase in tooth loss throughout life could be due to the mutilating characteristics of dental care, which historically, including in Brazil, fails to offer an alternative conservative treatment to tooth extraction [16, 21, 58–60]. In Brazil, although this situation has undergone positive transformations, the profile of oral health among Brazilian adults and elderly people has not yet changed [15, 17]. In fact, many people believe that edentulism is an inevitable outcome, a natural consequence of aging, a lack of self-care, which influences behaviors related to prevention and preservation of functional dentition [5, 16, 61].

Self-reported tooth loss was more prevalent in individuals who consumed more sugar at home. One study found that consuming sugary soft drinks more frequently increased the chances of tooth loss among young adults in the United States [62]. A systematic review pointed out that fermentable carbohydrates (sugars and starches) were the most common dietary risk factors for both dental caries and periodontal diseases, despite different associated mechanisms [63]. Dental caries is primarily caused by the interaction of biofilm with fermentable dietary carbohydrates on the tooth surface. Blood glucose generates oxidative stress and the advanced glycation end products can also trigger a hyperinflammatory state, causing periodontitis [63–65]. Tackling the excessive sugar consumption is now a dominant global public health priority. National and international nutritional guidelines now advocate a population-level reduction in sugar consumption. WHO recommends that children and adults reduce free sugar intake to less than 10% of total energy intake and that only 5% of total energy intake come from free sugars. In most countries in the world, however, sugar consumption is considerably higher than the WHO recommendation, especially among disadvantaged and low-income groups [66].

Dental pain, present in almost 20% of the sample, was also associated with a higher prevalence of tooth loss. In adults, dental pain affects daily activities, either functionally or socially [67]. The Brazilian last national oral health survey showed that the prevalence of dental pain within the previous six months was 27.5% in adults and 10.8% in elderly [9]. These numbers were similar for the North region (23.4% and 9.9%, respectively). Some reports have evidenced that dental pain is one of the main reasons for seeking dental care [18, 60, 68, 69], including in rural areas [2, 29, 53]. A study with elderly people living in a rural community of slave descendants in Brazil found that 62% of their last dental visit was due to pain/extraction [2]. Bhat et al. [53] showed that the main reasons for dentist appointments in a rural population in India were necessity, pain relief (31%), tooth extraction (54%), rather than regular or preventive dental care. Pain/urgent problem was responsible for almost 25% of clinical visits (while "hole tooth/fillings required" corresponded for 37%) among Aboriginal people attending rural and remote dental clinics, who presented a prevalence of 93.8% of dental pain in the previous six months [29]. As dental care is not easily accessible to people not living in capital city areas and care delivery models are often constrained, it is assumed that they are less willing to attend several dental appointments for preventive treatment and more likely to have a problem-orientated pattern of dental attendance. Many patients may wait until their dental problems become painful or until serious infections develop to seek for dental services areas [34, 70] and, for this reason, the incidence of dental extractions is high. In addition, dentists working in rural areas and small towns may have limited resources, little experience in providing specialized conservative dental treatment and so are less likely to supply preventive care than capital city dentists [34, 53]. In Brazil, the predominance of mutilating over conservative treatments is part of the recent history of public oral health, regardless of geographic location, as previously described.

Black or brown individuals reported less tooth loss. Although this issue is particularly complex in the study population [71], many studies report racial inequities in tooth loss. Most of

them point to a higher prevalence among blacks and browns [72, 73], although there is evidence that it may be higher among whites [59]. Gilbert et al. [74] put this question in perspective, stating that a comprehensive understanding of the total effect of race and socioeconomic status would need to take into account the effects of both. The characteristics related to the type of service used may also differ according to race/skin color, and should also be considered once racial inequalities go beyond socioeconomic differences [75].

Higher household income was an individual protective factor against self-reported tooth loss. A huge body of evidence supports this relationship between income and tooth loss in adults, whether in Brazilian [10, 15, 16, 20, 22, 25] or other countries populations [21, 26, 76], including two systematic reviews [77, 78]. The authors consider some possible explanations for this association, ranging from the structural to individual levels. Income disparity could represent a lack of investment in public resources such as dental services and water fluoridation, as the interests, needs and perceptions of the rich diverge from those of the poor. Furthermore, the presence of income inequality can lead to a non-cohesive society in which the dissemination of health information can be limited. Some studies have shown that low-income individuals are less likely to engage in preventive health behaviors, which play a relevant role in the establishment and progression of dental caries and periodontal disease. In addition to these factors, low-income individuals face more barriers to access dental services and economic restriction is also strongly associated with the type of dental treatment administered. While low-income individuals are more prone to tooth extraction, those with higher income are more likely to seek periodic consultations and conservative dental treatments, which results in the preservation of more teeth [10, 23, 74, 77, 79]. As in all oral diseases, there is a social relationship related to tooth loss: the lower income and schooling, the greater is tooth loss [10], depicting the pervasive inequality in this oral health outcome.

Cash transfer programs are important instruments for reducing family vulnerability [80] and they have shown a positive impact on the outcome related to oral health of individuals. The Bolsa Família program contributes to supplement household income, and the fulfillment of conditionalities guarantee food security, education, and health. Consequently, use of health facilities are also more frequent, including medical and dental appointments [81]. A family with better financial resources is also more able to overcome low availability and other barriers to accessing health services [39–41, 50]. Although a significant percentage of the population assessed are on the Bolsa Família program (44.6%), compared to the overall rates of the North region and Brazil, it could still be considered insufficient given the high rate of poverty and extreme poverty [82].

Individuals who attended dental appointments in the past year–almost half of the sample–reported a lower number of missing teeth. Studies carried out in the Brazilian population also showed that having had at least one dental appointment within this period was associated with a higher prevalence of favorable outcomes (preservation of functional dentition), which means that greater access to dental services is a factor associated with reduced tooth loss in Brazil [15, 20, 49, 83]. In accordance, other studies provided evidence that not having attended dental appointments in the past 12 months was associated with greater tooth loss [53, 73]. Dental appointment in the past 24 months has also proved to be a protective factor against tooth loss [16, 84].

Self-reported satisfaction with teeth and oral health was positively associated with reduced tooth loss, which is in agreement with previous findings [85]. This association is not surprising. The assessment of subjective measures improves the understanding of the consequences of oral diseases and tooth loss. Tooth loss causes functional impairment, for example, when chewing, and esthetic impairment, depending on where tooth loss occurred, which can ultimately affect the subjective perception of oral health and quality of life [6, 84].

There was no association between sex and self-reported tooth loss. Several studies have shown that this outcome is more frequent in women [11, 16, 59, 86], but this was not found in this population. Furthermore, contextual characteristics (per capita income, average household income, rate of poverty and extreme poverty, Gini index of per capita income and household income) were also not associated with tooth loss, as it might have been expected according to some studies. Although individual socioeconomic status has been associated with tooth loss, the less heterogeneous and highly vulnerable social context in these rural riverside locations [40] might have not allowed to identify differences in the assessed outcome.

Some limitations of this study include the cross-sectional design, thus causal inferences must be interpreted with caution. Considering the vulnerable situation of these populations, survival bias might be present. Information bias may also have occurred due to the self-reported data. Although the lack of a clinical examination to verify the oral health status of the adults interviewed could be considered a limitation, self-perception of the number of missing teeth has high validity according to the literature [87]. The questions used to assess tooth loss referred to the number of missing teeth, following the same criteria used in national health surveys carried out in Brazil [12]. However, despite the examiners having been trained, for individuals who have lost many teeth it can be difficult to remember exactly how many, tending to underestimate the number of missing teeth. Although more households than estimated in the sampling were included in the study due to the establishment of new families in some rural communities in the period between sampling and data collection, the minimum sample size was not achieved once the probability of finding residents in the households was lower than planned. As the sample size calculation considered the representativeness of the groups of interest and the analyzes were performed for adults and elderly together, the sample size remained representative for the study population. In addition, the prevalence of tooth loss in the study population was higher than the proportion used for the sample size calculation, which would require a smaller sample size than that calculated.

## Conclusions

The study population presented a high number of self-reported missing teeth. The presence of RHT was a contextual protective factor for tooth loss. As for individual factors, older age, white race/skin color, lower family income, non-inclusion in the Brazilian cash transfer program, higher sugar consumption, dental pain, worse self-perceived oral health and not having had a dental appointment over the past year were associated with a higher number of missing teeth. Tooth loss represents the endpoint of oral diseases and the failure of the dental healthcare model in preventing and controlling the most prevalent oral diseases. The high prevalence of this outcome found in individuals from rural riverside locations and the associated contextual and individual factors point at two directions: the need for prosthetic rehabilitation to restore functional dentition and esthetics and the need of a model of care that allows for preventive actions against tooth loss, which includes preventing main oral diseases, tooth caries and periodontal disease—the main causes of tooth loss. In addition, a broader approach in health promotion that addresses the social determinants of tooth loss, including intersectoral actions as well as individuals' empowerment and development of personal skills, especially considering the peculiar rural context, should be encouraged to promote effective action on the structural determinants to tackle the unrighteous but avoidable inequalities in rural oral health.

## Author Contributions

**Conceptualization:** Fernando José Herkrath, Luiza Garnelo, Andréia Coelho Gomes, Rosana Cristina Pereira Parente, Ana Paula Corrêa de Queiroz Herkrath.

**Data curation:** Vitor Guilherme Lima de Souza, Fernando José Herkrath, Andréia Coelho Gomes, Uriel Madureira Lemos, Rosana Cristina Pereira Parente, Ana Paula Corrêa de Queiroz Herkrath.

**Formal analysis:** Fernando José Herkrath, Ana Paula Corrêa de Queiroz Herkrath.

**Funding acquisition:** Luiza Garnelo, Rosana Cristina Pereira Parente.

**Investigation:** Vitor Guilherme Lima de Souza, Fernando José Herkrath, Luiza Garnelo, Andréia Coelho Gomes, Uriel Madureira Lemos, Ana Paula Corrêa de Queiroz Herkrath.

**Methodology:** Fernando José Herkrath, Luiza Garnelo, Andréia Coelho Gomes, Rosana Cristina Pereira Parente, Ana Paula Corrêa de Queiroz Herkrath.

**Project administration:** Luiza Garnelo, Rosana Cristina Pereira Parente.

**Resources:** Luiza Garnelo.

**Software:** Fernando José Herkrath.

**Supervision:** Fernando José Herkrath, Andréia Coelho Gomes, Uriel Madureira Lemos, Rosana Cristina Pereira Parente, Ana Paula Corrêa de Queiroz Herkrath.

**Validation:** Luiza Garnelo, Andréia Coelho Gomes, Rosana Cristina Pereira Parente, Ana Paula Corrêa de Queiroz Herkrath.

**Visualization:** Luiza Garnelo, Uriel Madureira Lemos.

**Writing – original draft:** Vitor Guilherme Lima de Souza, Ana Paula Corrêa de Queiroz Herkrath.

**Writing – review & editing:** Fernando José Herkrath, Luiza Garnelo, Andréia Coelho Gomes, Uriel Madureira Lemos, Rosana Cristina Pereira Parente.

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
