## [Decision Letter · Decision Letter 0]

15 Mar 2022

PONE-D-22-02459Contextual and individual factors associated with tooth loss among adults and elderly residents in rural riverside areasPLOS ONE

Dear Dr. Herkrath,

Thank you for submitting your manuscript to PLOS ONE. After careful consideration, we feel that it has merit but does not fully meet PLOS ONE’s publication criteria as it currently stands. Therefore, we invite you to submit a revised version of the manuscript that addresses the points raised during the review process. Please ensure that your decision is justified on PLOS ONE’s publication criteria and not, for example, on novelty or perceived impact.

We look forward to receiving your revised manuscript.

Kind regards,

João Gabriel Silva Souza

Academic Editor

PLOS ONE

Journal Requirements:

3. Please amend your current ethics statement to address the following concerns:

a) Did participants provide their written or verbal informed consent to participate in this study?

4. Thank you for stating the following in the Funding Section of your manuscript: 

"This study was funded by PROEP-Labs/ILMD Fiocruz Amazônia, call for proposals 001/2020."

We note that you have provided funding information. However, funding information should not appear in the Funding section or other areas of your manuscript. We will only publish funding information present in the Funding Statement section of the online submission form. 

"This study was funded by PROEP-Labs/ILMD Fiocruz Amazônia, call for proposals 001/2020 (https://amazonia.fiocruz.br). The funders had no role in study design, data collection and analysis, decision to publish, or preparation of the manuscript."

6. We note that Figure 1 in your submission contain map image which may be copyrighted. All PLOS content is published under the Creative Commons Attribution License (CC BY 4.0), which means that the manuscript, images, and Supporting Information files will be freely available online, and any third party is permitted to access, download, copy, distribute, and use these materials in any way, even commercially, with proper attribution. For these reasons, we cannot publish previously copyrighted maps or satellite images created using proprietary data, such as Google software (Google Maps, Street View, and Earth). For more information, see our copyright guidelines: http://journals.plos.org/plosone/s/licenses-and-copyright.

Reviewers' comments:

Reviewer's Responses to Questions

**Comments to the Author**

1. Is the manuscript technically sound, and do the data support the conclusions?

Reviewer #1: No

Reviewer #2: Yes

Reviewer #3: Yes

2. Has the statistical analysis been performed appropriately and rigorously? 

Reviewer #1: No

Reviewer #2: Yes

Reviewer #3: Yes

3. Have the authors made all data underlying the findings in their manuscript fully available?

Reviewer #1: Yes

Reviewer #2: Yes

Reviewer #3: Yes

4. Is the manuscript presented in an intelligible fashion and written in standard English?

Reviewer #1: Yes

Reviewer #2: Yes

Reviewer #3: Yes

5. Review Comments to the Author

Reviewer #1: Contextual and individual factors associated with tooth loss among adults and elderly residents in rural riverside areas

The objective of this study was to identify the association between individual and contextual socioeconomic and service-related factors as well as individual demographic, behavioral and subjective health factors for tooth loss among adults and elderly people living in rural riverside areas. The manuscript brings a result from a population few studied, wich is very relevant. However, some adjustment is necessary and also some points should be clarified.

Abstract:

the objective is too long, and individual is mentioned twice, that is repetitive.

The method should contain the period of the study. What is REDCap?

Why did the authors chose p-value ≤0.10 for the final model? Was the significance adopted 10%?

How was selected the sample?

Was there a sample size calculation?

Is “subjective health conditions” self-perception of oral health?

It is important to understand how were selected the variables.

Introduction

In my opinion tooth loss is not an indicator, but is a condition, the indicator is the prevalence of people who has experienced tooth loss or number of missing teeth, or others quantitative indicator of tooth loss.

I think when the authors describe the health services in North Region, it is important to contextualize the problem is not to be dependent of public health service, but there is a problem of accessibility for health. There is a lack in health service include private service. Besides Brazil have a universal health service, there are difficulties to manage the financial resources (lack of resources) and also is a too big country and unequal. I think these points could be in discussion section.

Methods

I didn´t understand why the sample calculation was 340 households, and the authors aimed to assess 851 individuals. Was the sample size calculation for households or individuals?

Also, I did not understand where this distribution “Of these, 294 were male adults, 258 were female adults, and 201 were elderlies aged 60 or older” came from. These data were expected or what the authors found?

I think there is a bias of information in the outcome. When the examinators ask for the number of missing teeth is more confused, than when is asked about the number of present teeth. Because, when people have lost many teeth is difficult to remember exactly how many.

I think can be underestimated this number of missing teeth.

How many interviewers have participated in the collection of data?

I think it is important the authors give more details of data collection, as how many questions were applied, how long was the interview, what time of the day was done the research. How was the pilot study? Was the questionnaire validated? How was evaluated sugar consumption?

The statistical analyses could be written in a different paragraph then data collection information.

For the outcome it is important to check: “the number of missing teeth, dichotomous outcomes of total tooth loss (complete edentulism), severe tooth loss (up to 8 natural teeth) and non-functional dentition (less than 20 natural teeth) were also described” severe tooth loss and non-functional dentition looks similar, because who has less than 20 natural teeth, can has up to 8 natural teeth. I think this variable could be better explained considering the number of missing teeth or present teeth in an ordinal way.

Average household income from contextual data, could be colinear with household income collected from the individual.

Why did the authors consider significance 10%? Is there a reference for this?

Results

The sample calculation was 340 households, why was examined 357 houses?

Table 2 is different than table 1 in terms of variable. What is the reference for sugar consumption?

And for household income?

I didn’t find the results commented in the conclusion.

Reviewer #2: Thank you for the opportunity to review the article and congratulations on the study. In general, the article is well described and methodologically adequate. However, I have a few points to consider.

In the 5th paragraph of the method, the authors classify as "poverty (family income less than US$ 1.90 per day) and extreme poverty (family income less than US$5.50 per day)", wouldn't it be the other way around?

In table 1, it is suggested that the currency to which the variables that include income refer be inserted in the footer.

I have a question about: if the variables were collected only once, why did the authors choose to express the incidence rate and not the prevalence, classically used for cross-sectional studies? Was it considered as a time interval the missing teeth during the life course? In addition, in the discussion the authors cite as "prevalence of tooth loss".

Reviewer #3: Title:

Include: `self-reported` tooth loss

- But if it fits, include the type of study.

Abstract:

Objective:

- Be direct, joining the two dots, as was done for contextual factorsL `between individual` / `as well as individual demographic, behavioral and subjective health factors `.

Include: `self-reported` tooth loss.

Methods

- Include: household-based.

- Was there a sample calculation? Is the study representative?

- When was the study carried out?

- What is the age group determined for adult? and elderly?

Results

- For the outcome of tooth loss, age is very important. What is the average age? and adults? and the elderly?

- `(non-functional dentition)`= but if you lose 12 it would no longer be. It was confusing to use this concept for 20 teeth and not functional (the result is loss and not maintenance).

- It would be important in the description to average the missing teeth by subgroup (adult and elderly).

- Make it more evident if individual or contextual factors are associated.

Introduction

- Although all topics well justify the study, it can be reduced.

-The objective of the introduction and abstract must be the same. Standardize.

Methods

- Do not abbreviate: `N. Sra.`.

- When was the study carried out?

- What is the age group determined for adult? and elderly?

- Split the paragraphs of contextual and individual variables. Even in addition to the variables, you could put the categories. Still, I ask you to follow this logic of always contextual and then individual for all data / results / tables. - inserted after explaining about the outcome categories.

- After theoretical and practical training, a pilot study was conducted in two rural communities, from territories other than those included in the main study. - Put in the paragraph on data collection.

- There is a conceptual error in the use of the terms `incidence` / `risk factor / `IRR` for a cross-sectional study. Make the correction for the entire study: `prevalence` /. `associated factor`.

- It could include a figure with the hierarchical model used.

Results

- The study did not reach the minimum sample size. Did it reach the minimum calculated by stratification of sex and age? No longer representative? Need to put on that.

- How many recruited? What is the response rate?

- For the outcome of tooth loss, age is very important. What is the average age? and adults? and the elderly?

- It would be important in the description to average the missing teeth by subgroup (adult and elderly).

- Make it more evident if individual or contextual factors are associated.

- `(non-functional dentition)`= but if you lose 12 it would no longer be. It was confusing to use this concept for 20 teeth and not functional (the result is loss and not maintenance).

- When you put the results of statistical tests, although they are widely used, you need to put them in the method, and not just in the result. Still, it is worth noting what the parameters would be for the reader's understanding.

- Include: `self-reported` tooth loss.

- In the analysis it would be important to change the reference category, so that the results do not show protection factor data. Confusing for discussion, greater chance and protection.

Tables

- Put `location and date` in all table captions.

- Include: `self-reported` tooth loss.

Table 1

- Legend: `study participants` - be more specific = adults and elderly

residents in rural riverside áreas.

Table 2 and 3

- Legend: incidence = review

- Legend: independent variables - be more specific = contextual and individual

- IRR = review / put in the footer

Discussion

- Include: `self-reported` tooth loss.

- Make it more evident if individual or contextual factors are associated.

- The discussion follows the logic of first discussing individual factors and then contextual factors. Why didn't the method and tables follow the same order?

- Before putting on the results of tooth loss, I could put a paragraph on the characteristics of the sample. Does it match reality? More women? brown? Half-life adult age.

- Avoid the use of numerical data that are equally repeated in the results topic.

- In comparison with other studies, the location, age group and year of collection need to be clear.

- Third paragraph: What does it help / dialogue in understanding your findings? Looks like what's in the intro. You need to relate to your results.

- risk factor = review

- There is a study of tooth loss in Brazil that can help in the interpretation of findings on age / use dental servisse and tooth loss: https://journals.plos.org/plosone/article/authors?id=10.1371/journal.pone.0219240

- The discussion of skin color will not resemble the context that has already been discussed about practices of access to mutilating services does not prevent tooth loss (topic of the type of health team). So, lack of access can keep teeth?

6. PLOS authors have the option to publish the peer review history of their article (what does this mean?). If published, this will include your full peer review and any attached files.

Reviewer #1: No

Reviewer #2: No

Reviewer #3: **Yes: **Manoelito Ferreira Silva Junior

---

## [Author Response · Author response to Decision Letter 0]

28 May 2022

We thank the reviewers for their comments and/or suggestions, which helped to clarify this work. We have numbered their comments for the authors to help us referring to them when needed. We shall deal with each comment here:

Journal Requirements:

Answer: PLOS ONE's style requirements were checked.

Answer: The complete questionnaire will be uploaded when submitting the revised version of manuscript.

3. Please amend your current ethics statement to address the following concerns:

a) Did participants provide their written or verbal informed consent to participate in this study?

Answer: The ethics statement was amended.

Methods, last paragraph

Removed: “...and the individuals signed an informed consent form to participate in the study.”

Added: “...and written informed consent was obtained from individuals to participate in the study.”

4. Thank you for stating the following in the Funding Section of your manuscript: 

"This study was funded by PROEP-Labs/ILMD Fiocruz Amazônia, call for proposals 001/2020."

We note that you have provided funding information. However, funding information should not appear in the Funding section or other areas of your manuscript. We will only publish funding information present in the Funding Statement section of the online submission form. 

"This study was funded by PROEP-Labs/ILMD Fiocruz Amazônia, call for proposals 001/2020 (https://amazonia.fiocruz.br). The funders had no role in study design, data collection and analysis, decision to publish, or preparation of the manuscript."

Answer: The funding-related text was removed from the manuscript text, as suggested. Amended statements were included within cover letter.

Answer: The authors are aware of the need to provide repository information for study data once the manuscript is accepted.

6. We note that Figure 1 in your submission contain map image which may be copyrighted. All PLOS content is published under the Creative Commons Attribution License (CC BY 4.0), which means that the manuscript, images, and Supporting Information files will be freely available online, and any third party is permitted to access, download, copy, distribute, and use these materials in any way, even commercially, with proper attribution. For these reasons, we cannot publish previously copyrighted maps or satellite images created using proprietary data, such as Google software (Google Maps, Street View, and Earth). For more information, see our copyright guidelines: http://journals.plos.org/plosone/s/licenses-and-copyright. We require you to either (1) present written permission from the copyright holder to publish these figures specifically under the CC BY 4.0 license, or (2) remove the figures from your submission:

Answer: The map in Figure 1 was prepared by a researcher from the study team, a member from the research support department of Leônidas and Maria Deane Institute, Oswaldo Cruz Foundation. The map in Figure 1 was prepared by a researcher from the study team, a member from the research support department of Leônidas and Maria Deane Institute, Oswaldo Cruz Foundation. Basemaps/shapefiles used are described below:

Municipal division data were obtained from Instituto Brasileiro de Geografia e Estatística, IBGE (public domain): https://www.ibge.gov.br/en/geosciences/territorial-organization/territorial-meshes/

Hydrography data were obtained from Instituto Nacional de Pesquisas Espaciais, INPE (public domain): http://www.dpi.inpe.br/Ambdata/English/

Such information was inserted in the figure caption of Fig 1 as requested.

-----

Reviewers' comments:

Reviewer #1:

The objective of this study was to identify the association between individual and contextual socioeconomic and service-related factors as well as individual demographic, behavioral and subjective health factors for tooth loss among adults and elderly people living in rural riverside areas. The manuscript brings a result from a population few studied, wich is very relevant. However, some adjustment is necessary and also some points should be clarified.

Abstract

1) the objective is too long, and individual is mentioned twice, that is repetitive.

Answer: The objective was rewritten to comply with the reviewer’s recommendation.

Abstract, Objective

Removed: “To identify the association between individual and contextual socioeconomic and service-related factors as well as individual demographic, behavioral and subjective health factors for tooth loss among adults and elderly people living in rural riverside areas.”

Added: “To identify the association of socioeconomic and service-related contextual factors and socioeconomic, demographic, behavioral and self-perceived oral health individual factors with self-reported tooth loss among adults and elderly people living in rural riverside areas.”

Introduction, last paragraph

Removed: “Thus, the aim of the study was to identify the role of socioeconomic, demographic and service-related factors in tooth loss among adults and elderly people living in rural riverside areas.”

Added: “Thus, the aim of the study was to evaluate the association of socioeconomic and service-related contextual factors and socioeconomic demographic, behavioral and self-perceived oral health individual factors with self-reported tooth loss among adults and elderly people living in rural riverside areas.”

2) The method should contain the period of the study. What is REDCap?

Answer: The period was added to the text (2019). The REDCap is now mentioned only in the body of the manuscript, being replaced in the abstract by “electronic forms”.

3) Why did the authors chose p-value ≤0.10 for the final model? Was the significance adopted 10%?

Answer: P-value ≤0.10 was the criteria used to retain the independent variable in the final model. The significance level adopted in the analyzes was 5%. This information has been added to the text.

Abstract, Method

Removed: “Variables with p-value ≤0.10 were kept in the final model.”

Added: “Variables with p-value ≤0.10 were kept in the final model and the significance level adopted was 0.05.”

Method, 6th paragraph

Added: “The significance level adopted was 0.05.”

4) How was selected the sample?

Answer: Considering the Abstract word limit, the information that the study sample was at random and representative for the study population was added. Detailing on the sampling strategies was presented in the Method section of the manuscript.

Abstract, Method

Removed: “Interviews were conducted using...”

Added: “Interviews were conducted in a representative random sample of dwellers aged ≥ 18 years, using...”

5) Was there a sample size calculation?

Answer: Yes, the sample size calculation was performed. This information was presented in the Methods section of the manuscript.

6) Is “subjective health conditions” self-perception of oral health?

Answer: We agree that the name used was very unspecific and we replaced it throughout the text with “self-perceived oral health”.

7) It is important to understand how were selected the variables.

Answer: The variables were selected according to the theoretical background presented in the introduction to the article, aiming to cover some of the main factors related to the outcome of interest. This information was added in the Methods section (5th paragraph).

Introduction

8) In my opinion tooth loss is not an indicator, but is a condition, the indicator is the prevalence of people who has experienced tooth loss or number of missing teeth, or others quantitative indicator of tooth loss.

Answer: We agree with the reviewer and the sentences using the word have been rewritten.

Abstract, Background

Removed: “Tooth loss is an epidemiological oral health indicator with high prevalence and negative impact on quality of life.”

Added: “Tooth loss is an oral health condition with high prevalence and negative impact on quality of life.”

Introduction, 1st paragraph

Removed: “Tooth loss is one of the main oral health indicators.”

Added: “Tooth loss is one of the main oral health problems.”

9) I think when the authors describe the health services in North Region, it is important to contextualize the problem is not to be dependent of public health service, but there is a problem of accessibility for health. There is a lack in health service include private service. Besides Brazil have a universal health service, there are difficulties to manage the financial resources (lack of resources) and also is a too big country and unequal. I think these points could be in discussion section.

Answer: The paragraph has been shortened and some of these points have been added to the discussion (3rd paragraph). Aspects related to the lower availability of services and access barriers were also emphasized.

Introduction, 5th paragraph

Removed: “The transfer of federal resources is lower than the national average, municipal management is complex, securing workers is difficult, and medium- and high-complexity health services are only offered in capital cities. To the limited provision and organization of healthcare of services clinics are associated low income, population dispersion and the large geographic distances typical of the Amazon region [39-41].”

Added: “To the limited provision and organization of healthcare of services clinics are associated low income, population dispersion, the large geographic distances and other barriers to access typical of the Amazon region [39-41].”

Methods

10) I didn´t understand why the sample calculation was 340 households, and the authors aimed to assess 851 individuals. Was the sample size calculation for households or individuals?

Also, I did not understand where this distribution “Of these, 294 were male adults, 258 were female adults, and 201 were elderlies aged 60 or older” came from. These data were expected or what the authors found?

Answer: The sample size calculation was performed for the individuals. Additionally, considering the dispersion of residents throughout the rural area, it was calculated how many households should be randomly selected to reach this sample size, considering the probability of finding individuals in the households (calculated from the average number of residents of the ages of interest in the households for each location). This selection of households was carried out in a systematic random manner, using sampling intervals for each rural locality. The quantitative presented for each group (adults of both sexes and the elderly) were calculated separately and adjusted for the size of the groups, considering that for the main study in which the study was inserted there was interest in specific conditions for each group. Considering the reviewer's statements, the paragraph has been rewritten to make it clearer and to eliminate this misunderstanding between the main study and that information inherent to this specific study.

Methods, 2nd paragraph

Removed: “Stratified random sampling was performed based on the number of individuals and households in each community, totaling 3811 people residing in 1350 households as reported by the CHW. The sample size calculation considered the representativeness of the groups of interest in the main project, adults and elderly of both sexes and children under the age of two years, and the probability of finding individuals from each group within households. The calculation considered a prevalence of 50% of the health outcomes of interest and precision of 0.05, in addition to 10% of possible losses or refusals, adjusted for the finite population, resulting in 340 households, aiming to assess 851 individuals of the target groups. Of these, 294 were male adults, 258 were female adults, and 201 were elderlies aged 60 or older.”

Added: “Stratified random sampling was performed based on the number of individuals and households in each community, totaling 3811 people residing in 1350 households as reported by the CHW. The sample size calculation considered a prevalence of 50% of the health outcomes of interest and precision of 0.05, in addition to 10% of possible losses or refusals, adjusted for the finite population, and was carried out in a representative way for adults of both sexes and the elderly, resulting in 753 individuals aged ≥ 18 years. Considering the probability of finding individuals from each group within households, a number of 340 households were estimated to be visited in the study, distributed throughout the territory according to the stratification of the sample by the population size of the forty rural localities comprised in the study.”

11) I think there is a bias of information in the outcome. When the examinators ask for the number of missing teeth is more confused, than when is asked about the number of present teeth. Because, when people have lost many teeth is difficult to remember exactly how many. I think can be underestimated this number of missing teeth.

Answer: The questions used in the study are the same questions that have been used in national health surveys carried out in the country (https://www.pns.icict.fiocruz.br/wp-content/uploads/2021/02/Questionario-PNS-2019.pdf, variables U23a and U24a). However, we agree with the reviewer's statement. Although some studies have demonstrated the validity of this self-reported measure, a population with higher occurrence of tooth loss may be more subjected to information bias. Therefore, this issue was included in the study limitations.

Discussion, 13th paragraph

Removed: “Information bias may also have occurred due to the self-reported data. Although the lack of a clinical examination to verify the oral health status of the adults interviewed could be considered a limitation, self-perception of the number of missing teeth has high validity according to the literature [86].”

Added: “Information bias may also have occurred due to the self-reported data. Although the lack of a clinical examination to verify the oral health status of the adults interviewed could be considered a limitation, self-perception of the number of missing teeth has high validity according to the literature [87]. The questions used to assess tooth loss referred to the number of missing teeth, following the same criteria used in national health surveys carried out in Brazil [12]. However, despite the examiners having been trained, for individuals who have lost many teeth it can be difficult to remember exactly how many, tending to underestimate the number of missing teeth.”

12) How many interviewers have participated in the collection of data?

Answer: Twenty examiners participated in the data collection (including professors and postgraduate students), who traveled on board on three trips so that it was feasible to cover the study population. This information was added to the manuscript text.

Methods, 4th paragraph

Added: “To make it feasible to cover the entire population dispersed in the study territory, twenty examiners were responsible for data collection.”

13) I think it is important the authors give more details of data collection, as how many questions were applied, how long was the interview, what time of the day was done the research. How was the pilot study? Was the questionnaire validated? How was evaluated sugar consumption?

Answer: The contextual and individual variables assessed were now described in detail and splitted into two paragraphs (Methods, paragraphs 5 and 6). Discussion about the validity of the outcome of interest (self-reported tooth loss) is presented in the Discussion section (study limitations). Information about the collection period, duration of the interview and regarding the pilot study are now presented in the Methods section (4th paragraph). As the study was part of a larger one, the number of questions varied according to the characteristics of the individuals included in the households (such as gender, age and diagnosis of a chronic disease), with an average application time of 60 minutes per household.

14) The statistical analyses could be written in a different paragraph then data collection information.

Answer: The paragraphs were reorganized as suggested by the reviewer.

15) For the outcome it is important to check: “the number of missing teeth, dichotomous outcomes of total tooth loss (complete edentulism), severe tooth loss (up to 8 natural teeth) and non-functional dentition (less than 20 natural teeth) were also described” severe tooth loss and non-functional dentition looks similar, because who has less than 20 natural teeth, can has up to 8 natural teeth. I think this variable could be better explained considering the number of missing teeth or present teeth in an ordinal way.

Answer: The number of missing teeth, a discrete numerical variable (count outcome) was the main outcome of interest for the study. The other dichotomous outcomes (edentulism, severe tooth loss and non-functional dentition) were also described to portray the rural riverine population tooth loss scenario more comprehensively. As noticed by the reviewer, these categories are not mutually exclusive. An edentulous individual certainly did not also have a functional dentition. These classifications of tooth loss have been widely used in studies on tooth loss and were reproduced in the same way in the present study. A sentence was added in the Methods section (7th paragraph) to make it clear that the conditions are not mutually exclusive.

Parker ML, Thornton-Evans G, Wei L, Griffin SO. Prevalence of and Changes in Tooth Loss Among Adults Aged ≥50 Years with Selected Chronic Conditions - United States, 1999-2004 and 2011-2016. MMWR Morb Mortal Wkly Rep. 2020 May 29;69(21): 641-646.

Ribeiro CG, Cascaes AM, Silva AE, Seerig LM, Nascimento GG, Demarco FF. Edentulism, Severe Tooth Loss and Lack of Functional Dentition in Elders: A Study in Southern Brazil. Braz Dent J. 2016 May-Jun;27(3): 345-52.

Peres MA, Lalloo R. Tooth loss, denture wearing and implants: Findings from the National Study of Adult Oral Health 2017-18. Aust Dent J. 2020 Jun;65 Suppl 1: S23-S31.

Kassebaum NJ, Bernabé E, Dahiya M, Bhandari B, Murray CJ, Marcenes W. Global Burden of Severe Tooth Loss: A Systematic Review and Meta-analysis. J Dent Res. 2014 Jul;93(7 Suppl): 20S-28S.

16) Average household income from contextual data, could be colinear with household income collected from the individual.

Answer: We do agree with the reviewer's statement. To test multicollinearity, we used the variance inflation factor (VIF) which showed that there is a severe correlation between average household income from contextual data and household income from the individual. Nonetheless, this was not identified between per capita income (contextual) and total household income (individual), variables included in the multiple regression model presented in Table 3. See below the Stata outputs.

 Variable | VIF 1/VIF 

-------------+----------------------

 logincome | 43.87 0.022797

 avg_income | 43.87 0.022797

-------------+----------------------

 Mean VIF | 43.87

 Variable | VIF 1/VIF 

-------------+----------------------

 logincome | 2.11 0.473706

 perc_income | 2.11 0.473706

-------------+----------------------

 Mean VIF | 2.11

17) Why did the authors consider significance 10%? Is there a reference for this?

Answer: The significance level adopted was 0.05. This information has now been added to the text (see comment #3).

Results

18) The sample calculation was 340 households, why was examined 357 houses?

Answer: As noticed, there were more households included in the study than estimated in the sampling. This occurred due to the establishment of new families in some rural communities in the period between sampling and data collection. Even so, the guidance for the field team was to use the previously defined sampling interval for the systematic selection of households, resulting in the inclusion of an additional house in some locations.

19) Table 2 is different than table 1 in terms of variable. 

Answer: The variable names are now standardized in the tables. The variables of dental health services utilization and satisfaction with teeth/oral health were dichotomized for the regression analyses, but they are described in detail in Table 1, as assessed.

20) What is the reference for sugar consumption? And for household income?

Answer: Continuous numeric variables have no reference category. The measure of association expresses the expected change in the exp(β) (rate ratio) of the outcome for each unit of increase in the independent variable.

21) I didn’t find the results commented in the conclusion.

Answer: The Conclusions section is now presented in the manuscript, encompassing the last paragraph of the discussion.

--------------

Reviewer #2:

Thank you for the opportunity to review the article and congratulations on the study. In general, the article is well described and methodologically adequate. However, I have a few points to consider.

1) In the 5th paragraph of the method, the authors classify as "poverty (family income less than US$ 1.90 per day) and extreme poverty (family income less than US$5.50 per day)", wouldn't it be the other way around?

Answer: We thank the reviewer for identifying the error. The values were inverted and are now corrected in the text.

2) In table 1, it is suggested that the currency to which the variables that include income refer be inserted in the footer.

Answer: As suggested by the reviewer, it was included.

3) I have a question about: if the variables were collected only once, why did the authors choose to express the incidence rate and not the prevalence, classically used for cross-sectional studies? Was it considered as a time interval the missing teeth during the life course? In addition, in the discussion the authors cite as "prevalence of tooth loss".

Answer: Poisson regression coefficients can be interpreted as the log of the rate ratio. Thus, we obtain the rate ratio by exponentiating the Poisson regression coefficient. This measure of association is traditionally referred as incidence rate ratio (IRR), but we agree with the reviewer that this terminology may not be appropriate due to the study design and for count outcomes. Therefore, we now named exp(β) as “rate ratios” throughout the study and explained it in the Methods section. Corrections were made in the text of the Method (6th paragraph) and Results (Tables 2 and 3) sections.

----------------

Reviewer #3:

Title:

1) Include: `self-reported` tooth loss

Answer: The title of the manuscript was modified as suggested by the reviewer, and so was the variable name throughout the whole text.

2) - But if it fits, include the type of study.

Answer: The title of the manuscript was modified as suggested by the reviewer.

Title

Removed: Contextual and individual factors associated with tooth loss among adults and elderly residents in rural riverside areas.

Added: Contextual and individual factors associated with self-reported tooth loss among adults and elderly residents in rural riverside areas: a cross-sectional household-based survey.

Abstract:

Objective:

3) Be direct, joining the two dots, as was done for contextual factorsL `between individual` / `as well as individual demographic, behavioral and subjective health factors `.

Include: `self-reported` tooth loss.

Answer: The objective was rewritten to comply with the reviewer’s recommendation.

Abstract, Objective

Removed: “To identify the association between individual and contextual socioeconomic and service-related factors as well as individual demographic, behavioral and subjective health factors for tooth loss among adults and elderly people living in rural riverside areas.”

Added: “To identify the association of socioeconomic and service-related contextual factors and socioeconomic, demographic, behavioral and self-perceived oral health individual factors with self-reported tooth loss among adults and elderly people living in rural riverside areas.”

Methods

4) Include: household-based.

Answer: It was included as suggested by the reviewer.

Abstract, Method

Removed: “A cross-sectional study was conducted…”

Added: “A cross-sectional household-based survey was conducted…”

5) Was there a sample calculation? Is the study representative?

Answer: Considering the Abstract word limit, information that the study sample was at random and representative for the study population was added. Detailing on the sampling strategies was presented in the Method section of the manuscript.

Abstract, Method

Removed: “Interviews were conducted using...”

Added: “Interviews were conducted in a representative random sample of dwellers aged ≥ 18 years, using...”

6) When was the study carried out?

Answer: The year in which the data collection was carried out was included in the text (2019).

7) What is the age group determined for adult? and elderly?

Answer: For the main study, adults were considered those aged ≥18 years and elderly ≥60 years. Thus, the present study included residents aged 18 or over, encompassing both groups (adults and elderly). This information was added in the Abstract and Method section (2nd paragraph) of the manuscript.

Results

8) For the outcome of tooth loss, age is very important. What is the average age? and adults? and the elderly?

Answer: We agree that there is a direct relationship between age and tooth loss. For this reason, age was included as an independent variable (continuous) in the modeling. The mean age was now included in the Abstract, and more detail is presented in the Results section (standard deviation and range). Subgroup analysis was not performed in the study.

9) `(non-functional dentition)`= but if you lose 12 it would no longer be. It was confusing to use this concept for 20 teeth and not functional (the result is loss and not maintenance).

Answer: The number of missing teeth, a discrete numerical variable (count outcome) was the main outcome of interest for the study. The other dichotomous outcomes (edentulism, severe tooth loss and non-functional dentition) were also described to portray the rural riverine population tooth loss scenario more comprehensively. This classification has been widely used in studies on tooth loss and was reproduced in the same way in the present study. A sentence was added in the Methods section (7th paragraph) to make it clear that the conditions are not mutually exclusive.

Parker ML, Thornton-Evans G, Wei L, Griffin SO. Prevalence of and Changes in Tooth Loss Among Adults Aged ≥50 Years with Selected Chronic Conditions - United States, 1999-2004 and 2011-2016. MMWR Morb Mortal Wkly Rep. 2020 May 29;69(21): 641-646.

Ribeiro CG, Cascaes AM, Silva AE, Seerig LM, Nascimento GG, Demarco FF. Edentulism, Severe Tooth Loss and Lack of Functional Dentition in Elders: A Study in Southern Brazil. Braz Dent J. 2016 May-Jun;27(3): 345-52.

Peres MA, Lalloo R. Tooth loss, denture wearing and implants: Findings from the National Study of Adult Oral Health 2017-18. Aust Dent J. 2020 Jun;65 Suppl 1: S23-S31.

Kassebaum NJ, Bernabé E, Dahiya M, Bhandari B, Murray CJ, Marcenes W. Global Burden of Severe Tooth Loss: A Systematic Review and Meta-analysis. J Dent Res. 2014 Jul;93(7 Suppl): 20S-28S.

10) It would be important in the description to average the missing teeth by subgroup (adult and elderly).

Answer: Subgroup analysis was not performed in the study (see comment #8).

11) Make it more evident if individual or contextual factors are associated.

Answer: The abstract has been modified according to the reviewer's suggestion.

Abstract, Results

Removed: “Tooth loss was greater in older individuals who had experienced dental pain over the past six months and whose sugar consumption was high. Black or brown individuals, individuals whose household income was higher, those who were on the Bolsa Família cash transfer program, those who consulted a dentist over the past year, those who reported satisfaction with their teeth/oral health, and those who lived in territories covered by riverside health teams reported less tooth loss.”

Added: “Contextual characteristic of primary healthcare offered was associated with the outcome. The tooth loss was lower in territories covered by riverside health teams. At individual level, tooth loss was greater in older individuals who had experienced dental pain over the past six months and whose sugar consumption was high. Black or brown individuals, individuals whose household income was higher, those who were on the Bolsa Família cash transfer program, those who consulted a dentist over the past year, and those who reported satisfaction with their teeth/oral health reported less tooth loss.”

Introduction

12) Although all topics well justify the study, it can be reduced.

Answer: In response to reviewer #1 suggestion, part of the introduction paragraph that dealt with services in the North region was removed, shortening the introduction. As the reviewer also highlighted the importance of evidencing the theoretical basis for choosing the variables, the other paragraphs were maintained.

13) The objective of the introduction and abstract must be the same. Standardize.

Answer: The objectives of the introduction and abstract were standardized, as suggested by the reviewer.

Methods

14) Do not abbreviate: `N. Sra.`.

Answer: It is now spelled out in full.

15) When was the study carried out?

Answer: The year in which the data collection was carried out was included in the text (2019).

16) What is the age group determined for adult? and elderly?

Answer: For the main study, adults were considered those aged ≥18 years and elderly ≥60 years. Thus, the present study included residents aged 18 or over, encompassing both groups (adults and elderly). There was no subgroup analysis. Information about age was actually missing from the text and it was added in the 2nd paragraph of Methods section.

17) Split the paragraphs of contextual and individual variables. Even in addition to the variables, you could put the categories. Still, I ask you to follow this logic of always contextual and then individual for all data / results / tables. - inserted after explaining about the outcome categories.

Answer: The paragraph was splitted and the manuscript was reorganized according to the logic suggested by the reviewer.

Methods

Removed: “Contextual variables included for each territory were per capita income, average household income, poverty (family income less than US$ 1.90 per day) and extreme poverty (family income less than US$ 5.50 per day) rates, Gini index of per capita income and household income. These variables were calculated based on information from the households evaluated in the territories. Individual characteristics were sex, age, race/skin color, household income, registration in the Bolsa Família program (Brazilian conditional cash transfer program for low-income families), occupation, monthly sugar consumption of family, time since last dental appointment, dental pain over the past six months and satisfaction with teeth/oral health.”

Added: “Independent variables were selected according to the theoretical background presented on the determinants of tooth loss. Contextual variables included for each territory were per capita income, average household income, poverty (family income less than US$ 5.50 per day) and extreme poverty (family income less than US$ 1.90 per day) rates, Gini index of per capita income and household income. These variables were calculated based on information from the households evaluated in the territories. Characteristic of primary healthcare was also assessed at a contextual level, including: 1) locations covered by fluvial health teams (FHT) and CHW, with health unit in the community, 2) locations covered by FHT and CHW, without health unit in the community, 3) locations only covered by CHW, and 4) locations covered by riverside health teams (RHT).

Individual characteristics were sex, age, race/skin color, household income, registration in the Bolsa Família program (Brazilian conditional cash transfer program for low-income families), occupation (yes/no), monthly sugar consumption of family, time since last dental appointment, dental pain over the past six months (yes/no) and satisfaction with teeth/oral health. Time since last dental appointment was evaluated according to the categories: over the past 12 months, more than 1 year up to 2 years ago, more than 2 years up to 3 years ago, more than 3 years ago, and never have been to the dentist. Satisfaction was assessed using a 5-point Likert scale, ranging from very satisfied to very dissatisfied. Sugar consumption was evaluated through the availability inferred by the frequency of monthly purchases of sugary foods at home.”

18) After theoretical and practical training, a pilot study was conducted in two rural communities, from territories other than those included in the main study. - Put in the paragraph on data collection.

Answer: The sentence was moved to the paragraph on data collection, as suggested by the reviewer.

19) There is a conceptual error in the use of the terms `incidence` / `risk factor / `IRR` for a cross-sectional study. Make the correction for the entire study: `prevalence` /. `associated factor`.

Answer: Poisson regression coefficients can be interpreted as the log of the rate ratio. Thus, we obtain the rate ratio by exponentiating the Poisson regression coefficient. This measure of association is traditionally referred as incidence rate ratio (IRR), but we agree with the reviewer that this terminology may not be appropriate due to the study design and for count outcomes. Therefore, we now name exp(β) as “rate ratios” throughout the study. Corrections were made in the text of the Method (6th paragraph) and Results (Tables 2 and 3) sections. 

The mention of the independent variable associated with the outcome in the study as a risk factor, identified in the Discussion section, was also corrected.

Discussion, 4th paragraph

Removed: “As expected, age was a risk factor for tooth loss.”

Added: “At individual level, age was associated with self-reported tooth loss, as expected.”

20) It could include a figure with the hierarchical model used.

Answer: The hierarchical model was now inserted in the manuscript, according to the logic presented below.

Methods

Added:

Figure 2. Hypothesized hierarchical analytical model for tooth loss in the rural riverside population.

Results

21) The study did not reach the minimum sample size. Did it reach the minimum calculated by stratification of sex and age? No longer representative? Need to put on that.

Answer: As noticed, there were more households included in the study (357) than estimated in the sampling (340). This occurred due to the establishment of new families in some rural communities in the period between sampling and data collection. Even so, the guidance for the field team was to use the previously defined sampling interval for the systematic selection of households, resulting in the inclusion of an additional house in some locations. 

On the other hand, the minimum sample size was not achieved as emphasized by the reviewer, certainly due to the lower probability of finding residents in the households than planned. As the sample size calculation considered the representativeness of the groups of interest (adults and elderly), this would only be a problem if the analyzes had been performed by subgroup. So, considering individuals over 18 years of age, the sample size is representative for the study population. In addition, the prevalence of tooth loss in the study population was higher than the proportion used for the sample size calculation (50%), which would require a smaller sample size than that estimated. This discussion was also included in the study limitations.

Discussion, 13th paragraph

Added: “Although more households than estimated in the sampling were included in the study, due to the establishment of new families in some rural communities in the period between sampling and data collection, the minimum sample size was not achieved once the probability of finding residents in the households was lower than planned. As the sample size calculation considered the representativeness of the groups of interest and the analyzes were performed for adults and elderly together, the sample size remained representative for the study population. In addition, the prevalence of tooth loss in the study population was higher than the proportion used for the sample size calculation, which would require a smaller sample size than that calculated.”

22) How many recruited? What is the response rate?

Answer: Refusals and losses were negligible in the study population. As explained in item #21, this was not the reason why the sample size initially planned had not been reached.

23) For the outcome of tooth loss, age is very important. What is the average age? and adults? and the elderly?

Answer: This issue was previously answered. Please see comment #8.

24) It would be important in the description to average the missing teeth by subgroup (adult and elderly).

Answer: Subgroup analysis was not performed in the study (see comment #8).

25) Make it more evident if individual or contextual factors are associated.

Answer: The paragraph has been rewritten in order to detail which contextual and individual variables were associated with the outcome.

Results, 3rd paragraph

Removed: “Table 3 shows the adjusted prevalence ratios, indicating the association of several contextual and individual variables with the outcome assessed. A higher number of missing teeth was found in older individuals, those who consumed more sugar at home and who had experienced dental pain over the past six months. Individuals who reported less tooth loss were those who declared their skin color as black or brown, those who benefit from the Bolsa Família program, those with a higher household income, those covered by the RHT, those who attended dental appointments over the past year, and those who reported satisfaction with their teeth/oral health.”

Added: “Table 3 shows the adjusted rate ratios, indicating the association of contextual characteristic of primary healthcare offered with self-reported number of missing teeth. Locations covered by riverine teams presented lower tooth loss than those covered by fluvial teams with CHW in a health facility in the territory. At individual level, a higher number of missing teeth was found in older individuals, those who consumed more sugar at home and who had experienced dental pain over the past six months. Individuals who reported less tooth loss were those who declared their skin color as black or brown, those who benefit from the Bolsa Família program, those with a higher household income, those who attended dental appointments over the past year, and those who reported satisfaction with their teeth/oral health.”

26) `(non-functional dentition)`= but if you lose 12 it would no longer be. It was confusing to use this concept for 20 teeth and not functional (the result is loss and not maintenance).

Answer: This issue was previously answered. Please see comment #9.

27) When you put the results of statistical tests, although they are widely used, you need to put them in the method, and not just in the result. Still, it is worth noting what the parameters would be for the reader's understanding.

Answer: Data analysis description has been improved in the Methods section.

Methods, 7th paragraph

“The data collected in the study were directly exported from REDCap to the database files of the Stata program. Initially, a descriptive analysis of the data was performed. In addition to the main outcome of interest, the number of missing teeth, dichotomous outcomes of total tooth loss (complete edentulism), severe tooth loss (up to 8 natural teeth) and non-functional dentition (less than 20 natural teeth) were also described. Then, Poisson regression analysis was performed to evaluate the variables associated with the number of missing teeth (count outcome), estimating the rate ratios and respective 95% confidence intervals. Poisson regression coefficients (β) can be interpreted as the log of the rate ratio. Thus, the rate ratio was obtained by exponentiating the Poisson regression coefficient. In the analyses, rate ratios represent the expected count outcome for X+1 divided by the expected count outcome for X. A multilevel modeling analysis was carried out to include the hierarchical structure or grouping of the study population in the respective territories. Variables with p-values lower than or equal to 0.20 in the bivariate analyses were included in the multiple analysis considering the hierarchy between individual and contextual variables in the model. The first model included only the contextual variables. The individual variables were included in the second model. Variables with p-values lower than or equal to 0.10 when included into the hierarchical model were maintained in the final model. The significance level adopted was 0.05.”

28) Include: `self-reported` tooth loss.

Answer: It was included as suggested by the reviewer.

29) In the analysis it would be important to change the reference category, so that the results do not show protection factor data. Confusing for discussion, greater chance and protection.

Answer: We fully agree with the reviewer, mainly for the individual characteristics. However, the authors faced challenges for solving this issue, which would involve using an unusual reference category for race/skin color and also using different references for the dichotomous categorical variables (for some variables, 'yes' and for others, 'no'), as can be seen in Table 3. In this way, the reference categories were maintained unchanged. But for the variable ‘Satisfaction with teeth/oral health’, the reference category was changed according to the reviewer's suggestion.

Tables

30) Put `location and date` in all table captions.

Answer: It was included as suggested by the reviewer.

31) Include: `self-reported` tooth loss.

Answer: It was included as suggested by the reviewer.

Table 1

32) Legend: `study participants` - be more specific = adults and elderly residents in rural riverside áreas.

Answer: It was modified as suggested by the reviewer.

Table 1, Title

Removed: “Contextual and individual characteristics of study participants.”

Added: “Contextual and individual characteristics of adults and elderly residents in rural riverside localities included in the study, Negro River, Manaus, Amazonas, 2019.”

Table 2 and 3

33) Legend: incidence = review

Answer: It was corrected, as explained in the response to the item #19.

34) Legend: independent variables - be more specific = contextual and individual

Answer: It was included as suggested by the reviewer.

35) IRR = review / put in the footer

Answer: It was corrected, as explained in the response to item #19.

Discussion

36) Include: `self-reported` tooth loss.

Answer: It was included throughout the text.

37) Make it more evident if individual or contextual factors are associated. The discussion follows the logic of first discussing individual factors and then contextual factors. Why didn't the method and tables follow the same order?

Answer: The discussion first addresses the contextual variable associated with the outcome (characteristic of primary healthcare offered, 3rd paragraph) and then the individual variables, aiming to respect the order presented in the Tables. More emphasis was given to this issue, as suggested by the reviewer.

38) Before putting on the results of tooth loss, I could put a paragraph on the characteristics of the sample. Does it match reality? More women? brown? Half-life adult age.

Answer: Aware of its relevance, the characterization of the sample was described in detail in the Results section. As the discussion is already too long, the authors opted to focus the Discussion section on the outcome of interest. Considering the sample design, the reality of the study population was reproduced, and the study has internal validity. 

39) Avoid the use of numerical data that are equally repeated in the results topic.

Answer: Numerical results that were repeated in the Discussion section have been removed (mainly in the second paragraph).

40) In comparison with other studies, the location, age group and year of collection need to be clear.

Answer: In response to the reviewer's previous suggestions, this information have been clarified in the manuscript.

41) Third paragraph: What does it help / dialogue in understanding your findings? Looks like what's in the intro. You need to relate to your results.

Answer: The authors have decided to remove this paragraph as it did not contribute to build a robust discussion on the outcome of interest and its associated factors.

42) risk factor = review

Answer: The entire manuscript text was revised, as suggested by the reviewer. The only three mentions of ‘risk factors’ maintained in the text refer to the theoretical background or other studies.

43) There is a study of tooth loss in Brazil that can help in the interpretation of findings on age / use dental servisse and tooth loss: https://journals.plos.org/plosone/article/authors?id=10.1371/journal.pone.0219240

Answer: The reference was inserted in the discussion of the study findings.

References

Added: 60. Silva Junior MF, Batista MJ, de Sousa MDLR. Risk factors for tooth loss in adults: A population-based prospective cohort study. PLoS One. 2019;14(7):e0219240. doi: 10.1371/journal.pone.0219240.

44) The discussion of skin color will not resemble the context that has already been discussed about practices of access to mutilating services does not prevent tooth loss (topic of the type of health team). So, lack of access can keep teeth?

Answer: Thank you for this comment. In fact, the study showed that the dental service utilization had a protective effect against tooth loss at individual level. Thus, other unassessed variables might compose the complex explanation of this finding related to race/skin color. The reference was replaced, and the paragraph has been rewritten to make it suitable.

Discussion, 7th paragraph

Removed: “Black or brown individuals reported less tooth loss. Although this issue is particularly complex in the study population [70], many studies report racial inequities in tooth loss. Most of them point to a higher prevalence among blacks and browns [71,72], although there is evidence that it may be higher among whites [59]. Gilbert et al. [73] put this question in perspective. With the small exception of dental self-extractions [74], which happens in rural areas, the only way to experience tooth loss is through dental care. Therefore, there may be social determinants of tooth loss that operate in opposite directions: people with worse socioeconomic conditions and black or brown people may be at lower risk of tooth loss because they are less likely to have access to dental care, but once they do have access, they are at greater risk for tooth loss. Thus, a comprehensive understanding of the total effect of race and socioeconomic status would need to take into account the effects of both, dividing the process into two steps [75].”

Added: “Black or brown individuals reported less tooth loss. Although this issue is particularly complex in the study population [71], many studies report racial inequities in tooth loss. Most of them point to a higher prevalence among blacks and browns [72,73], although there is evidence that it may be higher among whites [59]. Gilbert et al. [74] put this question in perspective, stating that a comprehensive understanding of the total effect of race and socioeconomic status would need to take into account the effects of both. The characteristics related to the type of service used may also differ according to race/skin color, and should also be considered once racial inequalities go beyond socioeconomic differences [75].”

References

Removed: “74. Gilbert GH, Duncan RP, Earls JL. Taking dental self-care to the extreme: 24-month incidence of dental self-extractions in the Florida Dental Care Study. J Public Health Dent. 1998; 58(2):131-4. doi: 10.1111/j.1752-7325.1998.tb02497.x. PMID: 9729757.”

Added: “75. Constante HM. Racial inequalities in public dental service utilization: Exploring individual and contextual determinants among middle-aged Brazilian adults. Community Dent Oral Epidemiol. 2020; 48(4):302-8. doi: 10.1111/cdoe.12533. PMID: 32237080.”

---

## [Decision Letter · Decision Letter 1]

20 Sep 2022

PONE-D-22-02459R1Contextual and individual factors associated with self-reported tooth loss among adults and elderly residents in rural riverside areas: a cross-sectional household-based surveyPLOS ONE

Dear Dr. Herkrath,

Thank you for submitting your manuscript to PLOS ONE. After careful consideration, we feel that it has merit but does not fully meet PLOS ONE’s publication criteria as it currently stands. Therefore, we invite you to submit a revised version of the manuscript that addresses the points raised during the review process.

We look forward to receiving your revised manuscript.

Kind regards,

Gaetano Isola, Ph.D.

Academic Editor

PLOS ONE

Journal Requirements:

Additional Editor Comments:

The authors should fully address all minor comments before the final assessment of the manuscript

Reviewers' comments:

Reviewer's Responses to Questions

**Comments to the Author**

1. If the authors have adequately addressed your comments raised in a previous round of review and you feel that this manuscript is now acceptable for publication, you may indicate that here to bypass the “Comments to the Author” section, enter your conflict of interest statement in the “Confidential to Editor” section, and submit your "Accept" recommendation.

Reviewer #1: All comments have been addressed

Reviewer #2: All comments have been addressed

2. Is the manuscript technically sound, and do the data support the conclusions?

Reviewer #1: Partly

Reviewer #2: Yes

3. Has the statistical analysis been performed appropriately and rigorously? 

Reviewer #1: Yes

Reviewer #2: Yes

4. Have the authors made all data underlying the findings in their manuscript fully available?

Reviewer #1: Yes

Reviewer #2: Yes

5. Is the manuscript presented in an intelligible fashion and written in standard English?

Reviewer #1: Yes

Reviewer #2: Yes

6. Review Comments to the Author

Reviewer #1: The authors work hard in the manuscript and letter to reply to the comments. Some points yet need to be discussed.

In the objective, I would like to read a shorter objective as: Objective: To identify contextual and individual factors with self-reported tooth loss among adults and elderly people living in rural riverside areas.

I also emphasize that when the authors describe the health services in North Region, it is important to contextualize the problem is not to be dependent of public health service, but there is a problem of accessibility for health. There is a lack in health service include private service. Besides Brazil have a universal health service, there are difficulties to manage the financial resources (lack of resources) and also is a too big country and unequal. I think these points could be in discussion section. Please this fact clear in introduction.

In methods when the authors mentioned 2019 means January 2019 until December 2019?

I did not understand the sample calculation and sample selection. Why was the sample calculation 751 and 340 were estimated to be visited? Maybe is better make clear that the mean of people in each house is 2, and in order to reach the sample size that is calculated for individuals, 340 houses were selected.

In the methods it is important to make clear the way the variables were analyzed, if count or noun variables and the category to avoid confusion.

Were the examiners calibrated?

I understand the explanation for the variable missing teeth, but it will be interesting to observe which category is more prevalent in this population, non-functional dentition? More than 8 teeth? Edentulism? If the sum of the categories is 100% it would be better to comprehend the result and the distribution of the interest variable. If they are not mutually exclusive you cannot observe the real distribution.

In discussion, what is more than a quarter? 25%?

The conclusion needs to be rewritten in order to be based on the main results and answer the objective. What are the associated factors of tooth loos in individual and contextual factors?

Reviewer #2: The answers were answered satisfactorily. The study has methodological quality and is relevant above all because it is in a poorly studied and culturally differentiated population. The results can contribute to the planning of specific policies for this population. Congratulations on the work.

7. PLOS authors have the option to publish the peer review history of their article (what does this mean?). If published, this will include your full peer review and any attached files.

Reviewer #1: **Yes: **Marília Jesus Batista de Brito Mota

Reviewer #2: No

---

## [Author Response · Author response to Decision Letter 1]

3 Oct 2022

We thank the reviewers for their comments and/or suggestions, which helped to improve the manuscript. We have numbered their comments to organize the responses. We shall deal with each comment here:

Reviewer #1

The authors work hard in the manuscript and letter to reply to the comments. Some points yet need to be discussed.

1) In the objective, I would like to read a shorter objective as: Objective: To identify contextual and individual factors with self-reported tooth loss among adults and elderly people living in rural riverside areas.

Answer: The study objective was rewritten, as suggested by the reviewer.

Abstract, Objective

Removed: “To identify the association between socioeconomic and service-related contextual factors and socioeconomic, demographic, behavioral and self-perceived oral health individual factors with self-reported tooth loss among adults and elderly people living in rural riverside areas.”

Added: “To identify contextual and individual factors associated with self-reported tooth loss among adults and elderly people living in rural riverside areas.”

Introduction, last paragraph

Removed: “Thus, the aim of the study was to evaluate the association of socioeconomic and service-related contextual factors and socioeconomic, demographic, behavioral and self-perceived oral health individual factors with self-reported tooth loss among adults and elderly people living in rural riverside areas.”

Added: “Thus, the aim of the study was to evaluate the association of contextual and individual factors with self-reported tooth loss among adults and elderly people living in rural riverside areas.”

2) I also emphasize that when the authors describe the health services in North Region, it is important to contextualize the problem is not to be dependent of public health service, but there is a problem of accessibility for health. There is a lack in health service include private service. Besides Brazil have a universal health service, there are difficulties to manage the financial resources (lack of resources) and also is a too big country and unequal. I think these points could be in discussion section. Please this fact clear in introduction.

Answer: Certainly, the issues raised are relevant to the understanding of access to health services in the North region. The low availability of services (public and private), added to geographic and financial barriers, and other social and organizational aspects (such as accommodation and acceptability), depicts a challenging scenario in such an unequal country. These aspects were addressed in the Introduction and in the Discussion section of the manuscript.

Introduction, 5th paragraph

Removed: “Contextual characteristics of geographic location contributes to inequalities between rural and urban areas, both in general and oral health. The urban environment can mitigate the negative aspects present in the rural environment, such as geographic barriers, socioeconomic deprivation, and limited access to health services [31-35]. Despite having a strong dependence on public healthcare services [36], the physical network of services is inadequate and health professionals are lacking in rural areas in Brazil [37]. The northern region of Brazil has the worst rates of use of health services in the country [38]. Health policies in the region lack institutional structure, continuity, and sensitivity to regional specificities. To the limited provision and organization of healthcare of services clinics are associated low income, population dispersion, the large geographic distances and other barriers to access typical of the Amazon region [39-41].”

Added: “Contextual characteristics of geographic location contributes to inequalities between rural and urban areas, both in general and oral health. The urban environment can mitigate the negative aspects present in the rural environment, such as geographic barriers, socioeconomic deprivation, and limited access to health services [31-35]. Brazil is a large and unequal country, and the northern region has the worst rates of use of health services in the country [38]. Despite having a strong dependence on public healthcare services [36], the physical network of public and private services is inadequate and health professionals are lacking in rural areas in Brazil [37]. Health policies in the region lack institutional structure, continuity, and sensitivity to regional specificities. In addition to the limited provision and organization of healthcare of services clinics, there are low income, population dispersion, the large geographic distances and other barriers to access typical of the Amazon region [39-41].”

Discussion, 3rd paragraph

Maintained: “This would contribute to a more rational use of the limited human, financial and structural resources characteristic of the northern region of Brazil and also to a greater resoluteness of health care [39-40].”

Discussion, 9th paragraph

Added: “A family with better financial resources is also more able to overcome low availability and other barriers to accessing health services [39-41,50].”

3) In methods when the authors mentioned 2019 means January 2019 until December 2019?

Answer: Information about the months has been added to the Methods section.

Methods, 1st paragraph

Removed: “A household-based cross-sectional survey was conducted in 2019 in rural riverside localities…”

Added: “A household-based cross-sectional survey was conducted from March to July 2019 in rural riverside localities…”

4) I did not understand the sample calculation and sample selection. Why was the sample calculation 751 and 340 were estimated to be visited? Maybe is better make clear that the mean of people in each house is 2, and in order to reach the sample size that is calculated for individuals, 340 houses were selected.

Answer: As suggested by the reviewer, the explanation was added in the paragraph of the sample calculation, in order to make it clearer.

Methods, 2nd paragraph

Removed: “Considering the probability of finding individuals from each group within households, a number of 340 households were estimated to be visited in the study…”

Added: “Considering the probability of finding individuals from each group within households (average of 2.215 adults per household), a number of 340 households were estimated to be visited in the study…”

5) In the methods it is important to make clear the way the variables were analyzed, if count or noun variables and the category to avoid confusion.

Answer: The dependent variable, number of missing teeth, was evaluated as a count outcome, as also described in the Methods section. Information on independent variables was added to the text, as suggested by the reviewer.

Methods, 5th and 6th paragraphs

Removed: Independent variables were selected according to the theoretical background presented on the determinants of tooth loss. Contextual variables included for each territory were per capita income, average household income, poverty (family income less than US$ 5.50 per day) and extreme poverty (family income less than US$ 1.90 per day) rates, Gini index of per capita income and household income. These variables were calculated based on information from the households evaluated in the territories. Characteristic of primary healthcare was also assessed at a contextual level, including: 1) locations covered by fluvial health teams (FHT) and CHW, with health unit in the community, 2) locations covered by FHT and CHW, without health unit in the community, 3) locations only covered by CHW, and 4) locations covered by riverside health teams (RHT).

Individual characteristics were sex, age, race/skin color, household income, registration in the Bolsa Família program (Brazilian conditional cash transfer program for low-income families), occupation (yes/no), monthly sugar consumption of family, time since last dental appointment, dental pain over the past six months (yes/no) and satisfaction with teeth/oral health. Time since last dental appointment was evaluated according to the categories: over the past 12 months, more than 1 year up to 2 years ago, more than 2 years up to 3 years ago, more than 3 years ago, and never have been to the dentist. Satisfaction was assessed using a 5-point Likert scale, ranging from very satisfied to very dissatisfied. Sugar consumption was evaluated through the availability inferred by the frequency of monthly purchases of sugary foods at home.”

Added: “Independent variables were selected according to the theoretical background presented on the determinants of tooth loss. Contextual variables included for each territory were per capita income, average household income, poverty (family income less than US$ 5.50 per day) and extreme poverty (family income less than US$ 1.90 per day) rates, Gini index of per capita income and household income. These continuous numeric variables were calculated based on information from the households evaluated in the territories. Characteristic of primary healthcare was also assessed at a contextual level using a nominal categorical variable, including: 1) locations covered by fluvial health teams (FHT) and CHW, with health unit in the community, 2) locations covered by FHT and CHW, without health unit in the community, 3) locations only covered by CHW, and 4) locations covered by riverside health teams (RHT).

Individual characteristics were sex (male/female), age (discrete numeric variable), race/skin color (white/black/brown/Asiatic/indigenous), household income (continuous numeric variable), registration (yes/no) in the Bolsa Família program (Brazilian conditional cash transfer program for low-income families), occupation (yes/no), monthly sugar consumption of family, time since last dental appointment, dental pain over the past six months (yes/no) and satisfaction with teeth/oral health. Time since last dental appointment was evaluated according to the categories: over the past 12 months, more than 1 year up to 2 years ago, more than 2 years up to 3 years ago, more than 3 years ago, and never have been to the dentist. Satisfaction was assessed using a 5-point Likert scale, ranging from very satisfied to very dissatisfied. Sugar consumption was evaluated in kilograms (continuous numeric variable) through the availability inferred by the frequency of monthly purchases of sugary foods at home.”

6) Were the examiners calibrated?

Answer: As the survey was conducted with closed-ended interview questions, the answers to which were reported by the participants, the examiners were trained, and a pilot study was carried out in rural communities not involved in the main study. No clinical measurement was performed in the study.

Methods, 4th paragraph

“After 40 hours of theoretical and practical training, a pilot study was conducted in two rural communities, from territories other than those included in the main study.”

7) I understand the explanation for the variable missing teeth, but it will be interesting to observe which category is more prevalent in this population, non-functional dentition? More than 8 teeth? Edentulism? If the sum of the categories is 100% it would be better to comprehend the result and the distribution of the interest variable. If they are not mutually exclusive you cannot observe the real distribution.

Answer: The number of missing teeth, a discrete numerical variable (count outcome) was the main outcome of interest for the study. The other dichotomous outcomes (edentulism, severe tooth loss and non-functional dentition) were also described to portray the rural riverine population tooth loss scenario more comprehensively. As described in the Methods section and noticed by the reviewer, these categories are not mutually exclusive, and for this reason they were not presented in this way. An edentulous individual certainly did not also have a functional dentition. These classifications of tooth loss have been widely used in studies on tooth loss and were reproduced in the same way in the present study. With mutually exclusive categories (if applicable), of the 27.4% who had non-functional dentition, excluding 12.1% of edentulous, the remaining 15.3% would be divided into severe tooth loss (7.5%) and non-functional dentition (7.8%). We emphasize that, even if not presented in the text, these values can still be calculated when evaluating the difference between the frequencies of the not mutually exclusive categories, in a sequential way (e.g., 27.4-19.6=7.8%, 19.6-12.1=7.5%).

Parker ML, Thornton-Evans G, Wei L, Griffin SO. Prevalence of and Changes in Tooth Loss Among Adults Aged ≥50 Years with Selected Chronic Conditions - United States, 1999-2004 and 2011-2016. MMWR Morb Mortal Wkly Rep. 2020 May 29;69(21): 641-646.

Ribeiro CG, Cascaes AM, Silva AE, Seerig LM, Nascimento GG, Demarco FF. Edentulism, Severe Tooth Loss and Lack of Functional Dentition in Elders: A Study in Southern Brazil. Braz Dent J. 2016 May-Jun;27(3): 345-52.

Peres MA, Lalloo R. Tooth loss, denture wearing and implants: Findings from the National Study of Adult Oral Health 2017-18. Aust Dent J. 2020 Jun;65 Suppl 1: S23-S31.

Kassebaum NJ, Bernabé E, Dahiya M, Bhandari B, Murray CJ, Marcenes W. Global Burden of Severe Tooth Loss: A Systematic Review and Meta-analysis. J Dent Res. 2014 Jul;93(7 Suppl): 20S-28S.

8) In discussion, what is more than a quarter? 25%?

Answer: 27.4% of adults had non-functional dentition. The intention was not to repeat the numerical value already presented in the Results section, but to draw attention to its expressiveness within the sample. In view of the reviewer's question, the sentence was rewritten.

Discussion, 2nd paragraph

Removed: “…and more than a quarter presented non-functional dentition.”

Added: “…and approximately 27% presented non-functional dentition.”

9) The conclusion needs to be rewritten in order to be based on the main results and answer the objective. What are the associated factors of tooth loos in individual and contextual factors?

Answer: The authors agree with the reviewer. When the last paragraph of the discussion was reorganized to become the conclusion during the review process, it ended up that way. So, the conclusion was now rewritten based on the study objectives and the main results, as well as pointing out at the end some implications of the study findings, as also suggested throughout the review process.

Conclusions

Added: “The study population presented a high number of self-reported missing teeth. The presence of RHT was a contextual protective factor for tooth loss. As for individual factors, older age, white race/skin color, lower family income, non-inclusion in the Brazilian cash transfer program, higher sugar consumption, dental pain, worse self-perceived oral health and not having had a dental appointment over the past year were associated with a higher number of missing teeth.”

Reviewer #2

The answers were answered satisfactorily. The study has methodological quality and is relevant above all because it is in a poorly studied and culturally differentiated population. The results can contribute to the planning of specific policies for this population. Congratulations on the work.

Answer: The authors are grateful for the reviewer's suggestions, which contributed to improving the manuscript.

---

## [Decision Letter · Decision Letter 2]

4 Nov 2022

Contextual and individual factors associated with self-reported tooth loss among adults and elderly residents in rural riverside areas: a cross-sectional household-based survey

PONE-D-22-02459R2

Dear Dr. Herkrath,

We’re pleased to inform you that your manuscript has been judged scientifically suitable for publication and will be formally accepted for publication once it meets all outstanding technical requirements.

Kind regards,

Gaetano Isola, Ph.D.

Academic Editor

PLOS ONE

Additional Editor Comments (optional):

Reviewers' comments:

Reviewer's Responses to Questions

**Comments to the Author**

1. If the authors have adequately addressed your comments raised in a previous round of review and you feel that this manuscript is now acceptable for publication, you may indicate that here to bypass the “Comments to the Author” section, enter your conflict of interest statement in the “Confidential to Editor” section, and submit your "Accept" recommendation.

Reviewer #1: All comments have been addressed

2. Is the manuscript technically sound, and do the data support the conclusions?

Reviewer #1: Yes

3. Has the statistical analysis been performed appropriately and rigorously? 

Reviewer #1: Yes

4. Have the authors made all data underlying the findings in their manuscript fully available?

Reviewer #1: Yes

5. Is the manuscript presented in an intelligible fashion and written in standard English?

Reviewer #1: Yes

6. Review Comments to the Author

Reviewer #1: Thank you for considering the points I have discussed. The manuscript Is very relevant and will bring more light for this population that is not well studied.

7. PLOS authors have the option to publish the peer review history of their article (what does this mean?). If published, this will include your full peer review and any attached files.

Reviewer #1: **Yes: **Marília Jesus Batista

---

## [Editor Report · Acceptance letter]

14 Nov 2022

PONE-D-22-02459R2 

Contextual and individual factors associated with self-reported tooth loss among adults and elderly residents in rural riverside areas: a cross-sectional household-based survey 

Dear Dr. Herkrath:

I'm pleased to inform you that your manuscript has been deemed suitable for publication in PLOS ONE. Congratulations! Your manuscript is now with our production department. 

Kind regards, 

on behalf of

Prof. Gaetano Isola 

Academic Editor

PLOS ONE